# Unlocking Volition: Proactive Intention Decoding via Interpretable Graph Learning of Multi-Region ECoG

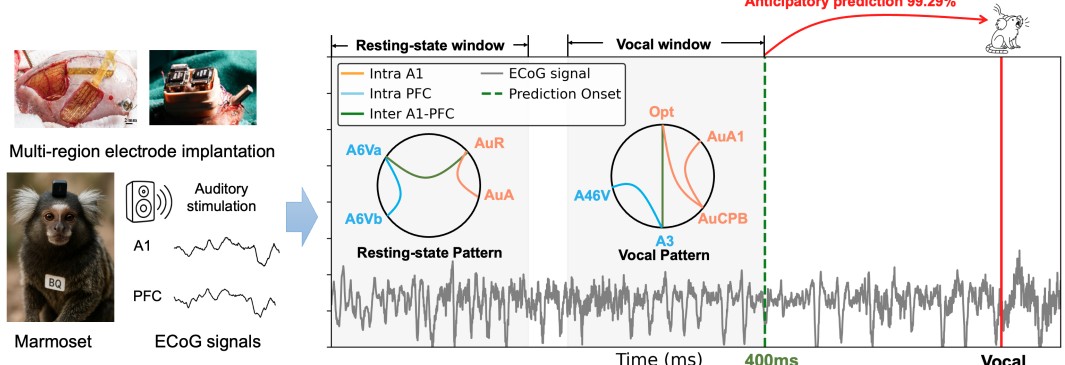

Figure 1: Overview of anticipatory marmoset vocalization prediction from multi-region ECoG recordings through ECoG-IBGT. Using common marmosets implanted with high-density electrode arrays across multiple cortical regions, we recorded high-fidelity ECoG signals from A1 and PFC under an auditory stimulation paradigm; our approach achieved 99.29% accuracy in predicting 400 ms before onset and revealed biologically meaningful patterns between vocalization and resting.

## Abstract

Current brain–machine interfaces (BMI) face fundamental limitations due to inherent latency from reliance on delayed motor cortical signals, and computational overhead, restricting their effectiveness in real-time applications such as rehabilitation therapy. Recent neuroscience indicates prefrontal and sensory cortical activities precede motor execution, offering an opportunity for proactive intent prediction. However, challenges remain in acquiring multi-region neural data, efficiently decoding high-dimensional signals, and ensuring model interpretability. To address these, we developed a high-density electrocorticography (ECoG)-based paradigm based on marmosets and introduced an information-bottleneck-driven graph transformer (ECoG-IBGT), reframing neural decoding as graph classification. Our method achieves 99.29% accuracy up to 400 ms before action onset with inherent interpretability, laying a foundation for reliable, low-latency BMIs. Code is available at [********************************URL Blinded for Review********************************].

## 1 Introduction

Human–computer interaction (HCI) is undergoing a profound transformation, with brain–machine interfaces (BMIs) increasingly envisioned as a pathway toward seamless and adaptive communication between humans and machines. Traditionally, most BMIs rely on decoding motor cortex activity to control external devices (Volkova et al., 2019). While motor–cortex–based decoding has achieved notable progress in both timeliness and accuracy, such approaches are fundamentally constrained: they capture the consequences of intention—motor execution—rather than intention itself (Tauste Campo et al., 2015; Sun et al., 2015; Ariani et al., 2022). As a result, current BMI systems

remain largely reactive, limiting their capacity for anticipatory, low-latency interaction (LaRocco & Paeng, 2020; Skomrock et al., 2018). To enable truly natural HCI, BMIs must move beyond "recognizing actions before they unfold" toward "identifying intentions at the moment they arise", thereby aligning interaction more closely with human cognitive mechanisms.

Recent neuroscientific evidence strongly motivates this paradigm shift. Studies in common marmosets have shown that upstream cortical regions, including the prefrontal cortex (PFC) and auditory cortex (AC), exhibit activation 200–300 ms prior to spontaneous vocalization (Mitelut et al., 2022; Tsunada & Eliades, 2025). These pre-vocal signals not only precede motor execution but also predict subsequent acoustic features, revealing a temporal window for proactive intention prediction. This suggests that intention-related information emerges in higher-order cortices before motor output, offering a compelling foundation for intention decoding. Thus we operationalize this evidence as a machine-learning task—Proactive Intention Decoding (PID)—that, given multi-regional neural windows at time $t$, predicts whether (and how) a behavior will occur at $t + \Delta$.

By reframing intention decoding as a machine learning problem, we aim to move BMIs from "post hoc recognition" to "proactive prediction", opening the door to the next generation of intention-aware, real-time human–machine interaction systems. Despite this insight, computational frameworks that can operationalize proactive intention decoding remain scarce. Current BMI technologies rarely exploit these upstream signals, and the gap between neuroscientific findings and machine learning practice is striking (Rouzitalab et al., 2023). Closing this gap introduces a new challenge for the ML community: how to design models that can capture weak, distributed, and dynamic neural patterns to predict internal intentions in real time:

(1) **High-quality, multi-regional neural acquisition**: Intention-related activity in PFC and AC is far weaker and more entangled than motor cortex signals (Sun et al., 2015; Ariani et al., 2022). Accurate decoding requires high-throughput recordings with broad coverage, yet EEG lacks resolution and implantable electrodes face biocompatibility limits. ECoG provides a promising compromise—good fidelity with moderate invasiveness—but current systems still suffer from limited coverage and durability, leaving a shortage of large-scale, high-quality datasets (Wang et al., 2023).

(2) **Scalable decoding of high-dimensional signals**: Most pipelines are tailored to low-dimensional EEG (Mitelut et al., 2022; Wu et al., 2016) and struggle with multi-regional ECoG. Hand-crafted features incur high costs and neglect dynamic inter-regional dependencies, limiting both accuracy and speed (Wang et al., 2016; Das et al., 2019). Early upstream signals are weak (Hockley et al., 2025b). Thus, new frameworks are needed that can extract discriminative patterns from high-dimensional ECoG, capture cross-region temporal–spectral dynamics, and enable real-time inference.

(3) **Limited interpretability**: While GNNs offer structural transparency, their explanations remain local or edge-level (e.g., GATs, GNNExplainer) (Veličković et al., 2018b; Ying et al., 2019). This hampers neuroscientific validity and reliability. Embedding interpretability during training—e.g., by isolating behaviorally relevant ensembles and motifs—could improve robustness, yield biologically meaningful insights into intent formation, and reduce the risk of failure in real-world BMIs.

To overcome these challenges, inspired by the vocalization prediction paradigm, we designed a vocalization-driven paradigm in common marmosets implanted with ultra-dense dual-region ECoG arrays (128 channels; 1.0–1.52 mm spacing) targeting primary auditory cortex (A1) and medial prefrontal cortex (mPFC). From this dataset, we propose ECoG-IBGT—an information-bottleneck-driven graph transformer (Dwivedi & Bresson, 2021) that reformulates neural decoding as graph classification. By integrating dynamic graph construction with a learnable IB subgraph generator, our approach achieves dimensionality reduction while preserving behaviorally-relevant multi-region dynamics. ECoG-IBGT reaches 99.29% prediction accuracy up to 400 ms before vocal onset, outperforming 12 benchmarks; an absolute gain of 5.63% over the strongest baseline (EEGNet at 93.66%). The extracted subgraphs reveal interpretable high-frequency pathways within and between A1 and PFC, implicating key auditory–prefrontal motifs in early intention encoding. Our framework bridges algorithmic performance and neuroscientific interpretability, providing a foundation for real-time ECoG decoding and low-latency brain–machine interfaces. To summarize, our key contributions are:

- **High-quality, multi-subject, multi-context ECoG dataset.** We collected a large-scale dataset by implanting ultra-dense dual-region ECoG arrays (128 electrodes, 1.0–1.52 mm spacing) in marmosets. Recordings spanned naturalistic paradigms including free movement and so-

cial interaction. The six curated datasets provide a rare, high-throughput resource for investigating spontaneous behavior and benchmarking early intention decoding in non-human primates—bridging a key translational gap between neuroscience and real-world BMI applications.

- **Accurate and proactive intention decoding architecture.** ECoG-IBGT reformulates neural decoding as a graph classification task, combining dynamic brain-graph construction with spatiotemporal attention. In the spontaneous vocalization task, it achieves 99.29% accuracy up to 400 ms before vocal onset—doubling the predictive horizon over existing models, providing a foundation for next-generation, low-latency BMI systems anticipating user intent in naturalistic settings, supporting neuroprosthetic control, speech decoding, and cognitive monitoring.

- **Built-in subgraph-level interpretability for neural decoding.** Our model introduces an information-bottleneck-driven subgraph generator that jointly constrains node and edge selection during training, yielding compact, behaviorally relevant subnetworks. These interpretable subgraphs uncover critical intra- and inter-regional communication patterns (e.g., A1–mPFC interactions), offering mechanistic insight into early volitional dynamics and establishing a scalable framework for cross-regional decoding in cognitive BMIs.

## 2 RELATED WORK

Decades of research have shown that multi-area recordings in primates can reveal temporally precise information flow during flexible behaviors, supporting distributed models of decision-making and sensorimotor integration Siegel et al. (2015). Recent work has underscored the need for naturalistic, wireless paradigms to capture the richness of multi-region dynamics in ethnologically relevant settings Cisek & Green (2024); Miller et al. (2022b).

The common marmoset offers a compelling model for such investigations, particularly in vocal behavior. Distributed circuits across auditory, pre-motor, and prefrontal regions orchestrate perception, planning, and context-dependent modulation of vocal output Roy & Wang (2016); Jovanovic et al. (2022); Zanini et al. (2023), forming a tractable framework for decoding spontaneous actions.

Although non-invasive EEG studies have reported moderate success in intention prediction using low-frequency movement-related potentials and data-driven classifiers (Bulea et al., 2014; Liu et al., 2018; Shafiul Hasan et al., 2020), ECoG approaches have demonstrated higher spatial-temporal resolution and much higher signal-to-noise ratio(SNR). For example, invasive ECoG has enabled continuous 2D trajectory decoding (Accuracy $\approx 0.74$) and grasp-force profile prediction (Accuracy $\approx 0.82$) in humans (Chen et al., 2014), and muscle activity estimation in marmosets with correlations up to $\approx 0.92$ (Shin et al., 2012; Umeda et al., 2019). ECoG studies have also revealed anticipatory oscillatory patterns in auditory tasks (Suda et al., 2022) and low-frequency waves following saccades in visual areas of the marmoset brain (Kaneko et al., 2022).

Despite recent advances, most ECoG studies remain confined to isolated brain regions or rely on highly controlled conditions(Miller et al., 2022a), which limits their ecological validity and interpretability. Multi-region recordings have begun to overcome these limitations. For instance, Betzel et al. constructed an inter-regional ECoG network model demonstrating that communication dynamics, anatomical geometry and gene co-expression can predict large-scale cortical connectivity (Betzel et al., 2017). Chronic wireless ECoG recordings in freely behaving marmosets have revealed that coordinated activity across auditory, premotor and prefrontal areas closely correlates with natural vocalisations and motor behaviour (Walker et al., 2021).

## 3 METHODS

### 3.1 PROBLEM DEFINITION

The aim of this study is to predict the marmosets' intention before vocalization. To facilitate mathematical modeling, we reformulated the problem as a graph classification task. Specifically, each marmoset's ECoG recordings consist of 128 channels. For the marmoset's ECoG time series of each channel, a $\delta$-second window preceding the vocalization and a window from the resting state are selected. For each vocalization event, we extract a single $\delta$-second, 128-channel window ending at 400 ms before onset; computing pairwise functional connectivity across the 128 channels

yields one graph. Rest trials are constructed analogously from non-overlapping baseline segments. (Fig. 2A). Each brain functional graph $G$ contains $n$ regions (channels), defined as $G = (X, A)$, where $X \in \mathbb{R}^{n \times n}$ are node features and $A \in \{0, 1\}^{n \times n}$ is an adjacency matrix, as shown in Fig. 2A. Thus, our goal is to train a neural network to learn a graph representation $h_G$ from the brain functional graph $G_{\text{ECoG}}$ to predict the binary vocalization label $y$:

$$f : G_{ECoG} \to h_G \to y_{vocal}, \quad y \in \{0, 1\}. \tag{1}$$

The overall framework of ECoG-IBGT is depicted in Fig 2B. We employ the concept of information bottleneck to achieve the dual objective of removing redundant information and accurately predicting the vocalization state. ECoG-IBGT comprises three main modules: IB-subgraph generator, graph encoder, and mutual information estimator. The IB-subgraph generator extracts a subgraph with soft edge masks $M_e$ and node masks $M_v$ from the original graph $G$. The graph encoder learns graph embeddings for both the $G$ and the subgraph $G_{\text{sub}}$. Then the mutual information (MI) $I(G; G_{\text{sub}})$ between $G$ and $G_{\text{sub}}$ was calculated.

## 3.2 IB-SUBGRAPH GENERATOR

Current brain network subgraph methods typically do not jointly constrain node and edge selection, resulting in subgraphs that are neither interpretable nor compact. To address this limitation, we propose a novel subgraph generator that simultaneously prunes redundant nodes and sparsifies inter-node connections, yielding concise and functionally meaningful substructures.

The procedure of the IB-Subgraph Generator is shown in Fig 2B. Specifically, we compute node mask $M_v$ and edge mask $M_e$:

$$M_v^{(i)} = \text{Gumbel-Softmax}(\sigma(\text{MLP}(x_i))), \quad M_e^{(ij)} = \text{Gumbel-Softmax}(\sigma(\text{MLP}(x_i; x_j))). \tag{2}$$

where $\sigma$ represents the sigmoid function and $x_i$, $x_j$ represent the node features.

To ensure the stability and robustness of the subgraph, we introduce the connectivity loss:

$$\mathcal{L}_{\text{conn}} = \left\| \text{Norm}(\hat{M}_v^{\top} M_e \hat{M}_v) - I_2 \right\|_F, \tag{3}$$

where $\hat{M}_v = [M_v, 1 - M_v]$.

We define $D = \hat{M}_v^{\top} M_e \hat{M}_v \in \mathbb{R}^{2 \times 2}$, and it has two important entries:

$$D_{11} = \sum_{i,j} \hat{M}_v^i M_e^{ij} \hat{M}_v^j, \quad D_{22} = \sum_{i,j} (1 - \hat{M}_v^i) M_e^{ij} (1 - \hat{M}_v^j). \tag{4}$$

where the size of $D_{11}$ quantifies connectivity within the subgraph, with a larger $D_{11}$ indicating stronger connectivity in its internal structure. $D_{22}$ represents connectivity outside the subgraph. Thus, minimizing connectivity loss preserves connectivity within the subgraph while promoting external sparsity, effectively removing redundant nodes and edges to yield a meaningful subgraph.

## 3.3 DUAL STRUCTURE GRAPH ENCODER

Dual structure graph encoder is used to obtain the graph embedding from the original graph and subgraph. We use the graph transformer as the backbone network. In addition, we employ an interpretability-driven message passing mechanism to update the node embedding. The process of obtaining node embeddings in a traditional graph transformer can be expressed as follows:

$$h_v^{(\ell+1)} = \text{ReLU}\Big(\text{BN}\big(\text{TFConv}(h_v^{(\ell)}, E)\big)\Big), \quad \text{TFConv}(h_v^{(\ell)}, E) = \sum_{k \in \mathcal{N}(v)} \alpha_{vk} \cdot W h_v. \tag{5}$$

where BN denotes batch normalization layer, TFConv represents graph transformer layer, $\alpha_{vk}$ is the attention weight, and $W$ is the weight matrix used for value transformation.

To ensure interpretability and stability, we design an explainable message propagation architecture that seamlessly integrates explanation signals including node mask and edge mask during model training. Given node mask $M_v$ and edge mask $M_e$, the message passing process is defined as:

$$h_v^{(\ell+1)} = \text{ReLU}\Big(\text{BN}\big(\text{TFConv}(h_v^{(\ell)}, E, M_e)\big)\Big) \odot M_v. \tag{6}$$

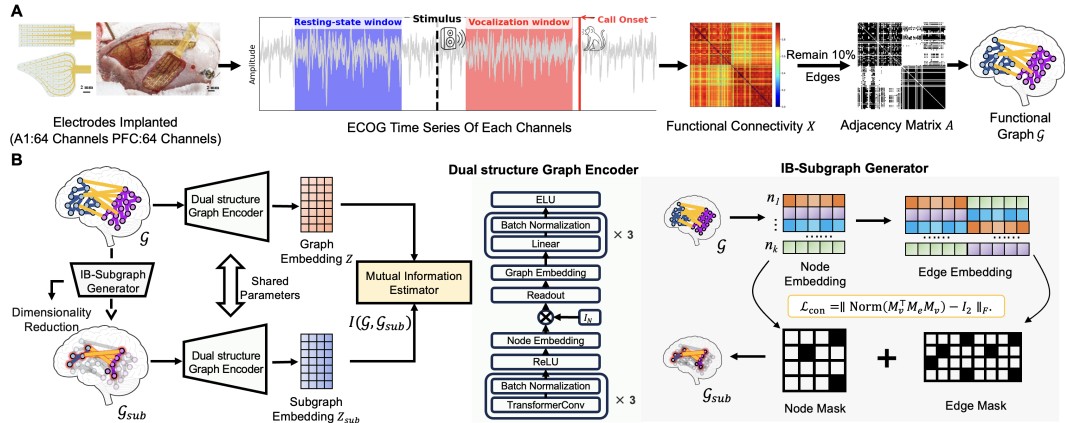

Figure 2: Overview of the graph-construction pipeline and proposed ECoG-IBGT framework. (A) ECoG signals are segmented into a resting-state window and a vocalization-state window. We compute pairwise Pearson's correlations across channels to obtain the functional connectivity matrix $X$. To construct the adjacency matrix $A$, we retain the top 10% of edges from $X$, yielding the functional graph $G = (A, X)$. (B) The original graph $G$ is pruned by the IB-Subgraph Generator to produce the subgraph $G_{\text{sub}}$. Graph embeddings for both $G$ and $G_{\text{sub}}$ are generated via the Dual Structure Graph Encoder. Finally, we estimate the MI $I(G, G_{\text{sub}})$ using a mutual-information estimator.

### 3.4 MUTUAL INFORMATION ESTIMATOR

The graph IB objective includes two MI terms:

$$\min_{G_{\text{sub}}} I\left(G_{\text{sub}}, G\right) - \beta I\left(G_{\text{sub}}, y\right). \tag{7}$$

Minimizing $-I\left(\mathcal{G}_{\text{sub}}, y\right)$ (a.k.a., maximizing $I\left(\mathcal{G}_{\text{sub}}, y\right)$) encourages $G_{\text{sub}}$ is most predictable to graph label $y$. Mathematically, we have:

$$-I(G_{\text{sub}}, y) \leq \mathbb{E}_{y, G_{\text{sub}}}[-\log q_\theta(y|G_{\text{sub}})] := \mathcal{L}_{\text{CE}}(G_{\text{sub}}, y). \tag{8}$$

where $q_\theta\left(y|G_{\text{sub}}\right)$ is the variational approximation of the true mapping $p\left(y|G_{\text{sub}}\right)$ from $G_{\text{sub}}$ to $y$. Eq. (8) indicates that minimizing $-I\left(G_{\text{sub}}, y\right)$ approximately equals minimizing the cross-entropy loss $\mathcal{L}_{\text{CE}}$.

As for the MI term $I\left(G_{\text{sub}}, G\right)$, we use Hilbert-Schmidt independence criterion (HSIC) to calculate the mutual information between $G$ and $G_{\text{sub}}$. HSIC directly estimates the MI without the need for discrete probability density function (PDF) estimation or any auxiliary neural networks. Formally, $I\left(G_{\text{sub}}, G\right)$ can be defined as:

$$I\left(G_{\text{sub}}, G\right) = (N-1)^{-2}\text{tr}(K_G H K_{G_{\text{sub}}} H), \tag{9}$$

where $N$ is the number of batch size, tr is the trace operator, $K_G$ and $K_{G_{\text{sub}}}$ are kernel matrices with $K_G^{ij} = k_G(Z^i, Z^j)$ and $K_{G_{\text{sub}}}^{ij} = k_{G_{\text{sub}}}(Z_{\text{sub}}^i, Z_{\text{sub}}^j)$, respectively and $H \in \mathbb{R}^{N \times N}$ is the centering matrix $H = I_{ij} - \frac{1}{N}$, $Z$ and $Z_{\text{sub}}$ represent the node embeddings derived from the graph encoder.

In this work, we use the radial basis function (RBF) kernel $\kappa$ to obtain $k_G$ and $k_{G_{\text{sub}}}$:

$$\kappa\left(z^i, z^j\right) = \exp\left(-\frac{\left\|z^i - z^j\right\|^2}{2\sigma^2}\right). \tag{10}$$

For the kernel width $\sigma$, we estimate 10 nearest distances of each sample and obtain the mean. We set the $\sigma$ with the average of mean values for all samples.

Thus, the final objective of ECoG-IBGT is:

$$\mathcal{L} = \mathcal{L}_{\text{CE}} + \alpha\,\mathcal{L}_{\text{conn}} + \beta\,I\left(G_{\text{sub}}, G\right). \tag{11}$$

where $\alpha$ and $\beta$ balance the contributions of each component.

## 4 EXPERIMENTS

This section evaluates the effectiveness and presents the interpretability of ECoG-IBGT with extensive experiments. We aim to address the following research questions:

**RQ1.** Can the high-SNR, high spatiotemporal–resolution ECoG signals acquired by our multi-region paradigm enable earlier and more accurate intention prediction?

**RQ2.** How does ECoG-IBGT perform compared with state-of-the-art models of various types?

**RQ3.** What interpretable neural mechanisms does ECoG-IBGT provide to guide dimensionality reduction and enable anticipatory prediction?

### 4.1 EXPERIMENTAL SETTINGS

**Datasets.** Leveraging two adult common marmosets (named BQ and AZ) implanted with ECoG electrode (128 electrodes, 1.0–1.52 mm spacing) covering A1 (64 channels) and PFC (64 channels), we employed a series of paradigms to achieve dense ECoG recordings. Signals were sampled at 500 Hz and band-pass filtered (1–200 Hz) to remove power-line and high-frequency noise. Table 1 summarizes the paradigm settings. For each marmoset, sessions correspond to pure-tone stimulation (i.e., 2,000, 4,000, 8,000, and 16,000 Hz), conspecific-call stimulation (i.e., playback of recorded marmoset vocalizations), and antiphonal-calling interactions. Each session lasted approximately 32 minutes, yielding a total of 233 minutes of recordings across all sessions. To balance the requirements of system latency and behavioral responsiveness, a 400 ms margin before vocal onset provided lead time for downstream execution, while also ensuring that data acquisition, feature extraction, and algorithmic inference can be completed in time to support real-time, closed-loop brain–machine interaction. Our data acquisition has passed the ethical certification of relevant institutions.

| Stimulus | Subject BQ | Subject AZ |
|---|---|---|
| Pure-tone recordings | B1-B4 | – |
| Phee recordings | – | A1 (chair-fixed), A2 (free-moving) |
| Antiphonal calling | B5-B7 | A3 |

Table 1: Sessions across paradigms. B1–B7 and A1–A3 are session identifiers for the recordings.

Based on the experimental subject, stimulation paradigm, and the marmoset's behavioral state, we assembled 6 datasets (Trial-1 through Trial-6):

- **Trial-1**: pooled 7 sessions (B1–B7) from BQ, yielding 770 trials (385 vocalization vs. 385 non-vocalization);

- **Trial-2**: comprised 3 sessions (A1–A3) from AZ, also totaling 770 trials (385/385);

- **Trial-3**: combined 4 sessions (B5–B7 and A3) in which antiphonal calling by paired marmosets elicited vocalizations, resulting in 1012 trials (506/506);

- **Trial-4**: merged 6 sessions (B1–B4 with pure-tone and conspecific-call playback, plus A1–A2), likewise producing 1 012 trials (506/506);

- **Trial-5**: consisted of a single session (A2) under free-behavior conditions with playback-induced stimuli, containing 222 trials (111/111);

- **Trial-6**: comprised 1 session (A1) under chair-restrained conditions with identical playback stimuli, also yielding 222 trials (111/111).

**Cross-Session Evaluation.** The robustness of our method under distribution shifts were further evaluated by performing cross-session decoding across different trial groups (see Appendix H).

**Metrics.** We evaluate all methods using the following metrics: Accuracy, Precision, Recall, F1, AUC-ROC, AUC-PR and inference time per sample (ms). All reported performances are the average of 5 random runs on the test set.

**Implementation Details.** For experiments, we use a three-layer graph transformer module and set the number of heads $M$ to 3 for each layer. We randomly split the data into 70% for training, 10% for validation, and the remaining 20% as the test set. In the training process of our proposed model, we use an Adam optimizer with an initial learning rate of $5 \times 10^{-5}$, a weight decay of $1 \times 10^{-5}$ and set the loss-balancing hyperparameters $\alpha$ and $\beta$ both to $1 \times 10^{-5}$. The batch size is set to 128. All models are trained for 150 epochs, and the epoch with the highest accuracy on the validation set is selected for performance comparison on the test set. The model is trained on an NVIDIA H20-NVLink. Please refer to the repository for the full implementation of ECoG-IBGT.

## 4.2 MULTI-REGION PARADIGM DECODING (RQ1)

**Temporal Influence on Predictive Anticipation.** As shown in subfigure (a) of Figure 3, we compare the decoding accuracy of multi-region vs. single-region ECoG signals using a sliding 3 s window from $-1$ s to $-400$ ms relative to vocal onset (where $-1$ s and $-400$ ms denote the lead time at the window's end). The multi-region model outperforms single-region models at almost all anticipation time points, with the greatest gain observed between $-800$ ms and $-550$ ms. Subfigure (b) confirms this trend across 6 independent datasets.

**Validation Across 6 Datasets.** To validate reliability, we compare decoding performance across 6 sessions using multi-region vs single-region ECoG signals, as shown in subfigure (b) of Figure 3. The advantage of multi-region decoding is robust, consistently observed across datasets, with pronounced improvements in decoding accuracy over traditional auditory cortical regions.

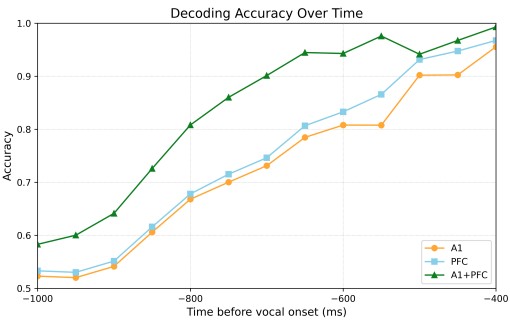
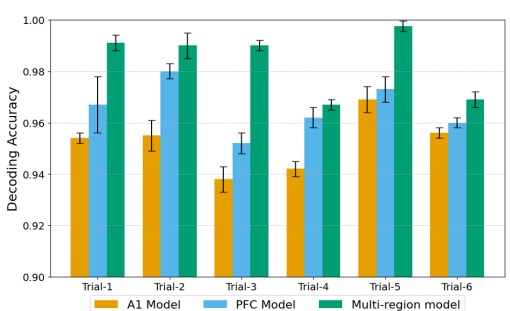

(a) Prediction accuracy vs. lead time for A1, PFC, and multi-region models.

(b) Decoding accuracy (with standard deviation) for A1, PFC, and multi-region across 6 sessions.

Figure 3: (a) Temporal anticipation: decoding performance as a function of lead time. (b) Session-wise comparison across 6 datasets of single- vs multi-region models.

## 4.3 PERFORMANCE EVALUATION (RQ2)

**Comparative Study.** We compare ECoG-IBGT with baselines of 4 types (see Table 2). ECoG-IBGT achieves the highest accuracy (99.29%), F1 (99.31%), AUC-ROC (99.99%) and AUC-PR (99.99%). **(a) ECoG-IBGT vs. GNN baselines.** ECoG-IBGT achieves more than 14% absolute improvements over GNN baselines such as GIN (Xu et al., 2019), GAT (Veličković et al., 2018a), GCN (Kipf & Welling, 2017) and GraphSAGE (Hamilton et al., 2017).**(b) ECoG-IBGT vs. temporal prediction model.** We compare two temporal models, EEGNet (Lawhern et al., 2018) and MedFormer (Wang et al., 2024). ECoG-IBGT achieves a 5.63% improvement over EEGNet (paired t-test, $p < 0.001$) and nearly 39% over MedFormer. **(c) ECoG-IBGT vs. BrainIB** (Zheng et al., 2024). BrainIB also employs the information bottleneck method for brain network analysis. Our method achieves significantly higher accuracy than BrainIB, demonstrating the effectiveness of our unique design tailored to ECoG signals. **(d) ECoG-IBGT vs. graph-based dimensionality reduction methods.** We compare 5 graph-based dimensionality reduction methods including TopKPool (Cangea et al., 2018), DiffPool (Ying et al., 2018), SAGPool (Lee et al., 2019), Cluster-GT (Huang et al., 2024), and GraphPCA (Yang et al., 2024). The superiority of ECoG-IBGT against these methods suggests the effectiveness of dimensionality reduction. Moreover, we observed that by reformulating the task from temporal-sequence prediction to graph classification, ECoG-IBGT

achieves at least a 40% reduction in inference time compared to temporal models. To further validate generalization, we conducted additional experiments on the BCI Competition III Dataset I benchmark. The corresponding results are reported in Appendix G.

| Model | Accuracy | Precision | Recall | F1-score | AUC-ROC | AUC-PR | Time (ms) |
|---|---|---|---|---|---|---|---|
| GIN | 84.64 ±6.47 | 84.38 ±11.41 | 90.25 ±11.01 | 86.02 ±5.71 | 93.39 ±6.09 | 92.81 ±8.21 | 0.217 |
| GAT | 81.43 ±1.12 | 94.05 ±0.25 | 69.12 ±1.87 | 78.18 ±1.32 | 94.69 ±0.31 | 94.90 ±0.28 | 0.304 |
| GCN | 79.87 ±0.89 | 96.30 ±0.18 | 63.73 ±2.05 | 76.68 ±1.45 | 96.66 ±0.15 | 97.22 ±0.12 | 0.173 |
| GraphSAGE | 85.13 ±1.11 | 98.46 ±4.70 | 71.94 ±4.59 | 82.89 ±1.48 | 96.62 ±1.85 | 97.02 ±2.00 | 0.181 |
| BrainIB | 78.57 ±0.42 | 73.21 ±1.98 | 91.28 ±0.52 | 81.20 ±0.87 | 90.22 ±0.67 | 91.21 ±0.55 | 3.308 |
| EEGNet | 93.66 ±0.31 | 93.76 ±0.26 | 93.65 ±0.32 | 93.65 ±0.32 | 98.04 ±0.12 | 98.47 ±0.09 | 0.619 |
| MedFormer | 60.48 ±1.98 | 59.90 ±2.15 | 59.55 ±2.22 | 59.53 ±2.17 | 61.84 ±1.89 | 61.20 ±1.95 | 0.604 |
| TopKPool | 89.29 ±0.67 | 93.92 ±0.28 | 83.95 ±0.92 | 88.02 ±0.65 | 97.10 ±0.13 | 96.98 ±0.15 | **0.089** |
| DiffPool | 79.22 ±1.33 | 78.63 ±1.47 | 82.52 ±1.25 | 80.49 ±1.38 | 87.62 ±0.82 | 87.93 ±0.76 | 0.096 |
| SAGPool | 90.68 ±0.91 | 94.18 ±0.26 | 87.31 ±0.85 | 90.48 ±0.72 | 95.20 ±0.34 | 96.75 ±0.22 | 9.827 |
| Cluster-GT | 81.07 ±1.29 | 80.34 ±2.85 | 80.54 ±3.12 | 80.35 ±1.22 | 89.37 ±1.22 | 87.94 ±1.51 | 0.524 |
| GraphPCA | 66.27 ±1.21 | 68.44 ±1.89 | 66.85 ±1.77 | 65.72 ±1.94 | 79.29 ±1.25 | 82.57 ±1.12 | 4.614 |
| **ECoG-IBGT** | **99.29** ±0.30 | **99.88** ±0.02 | **98.75** ±0.07 | **99.31** ±0.04 | **99.99** ±0.01 | **99.99** ±0.01 | 0.362 |

Table 2: Performance vs. standard deviations (%). **Bold**: best; Underlined: second best.

**Cross-subject generalization.** To further assess whether ECoG-IBGT captures behavior-relevant motifs beyond a single individual, we evaluate cross-subject decoding where training and test sets are strictly separated by subject. Specifically, we train on all graphs from AZ and test on BQ (AZ→BQ). As shown in Table 3, cross-subject decoding is substantially more challenging than the within-subject setting, yet ECoG-IBGT still achieves the best overall performance among all baselines (e.g., $72.13 \pm 1.50$ accuracy and $77.53 \pm 1.57$ AUC-ROC in AZ→BQ). These results indicate that the learned subgraphs are not merely subject-specific artifacts but capture transferable, behavior-relevant multi-region motifs.

| Model | Acc | Prec | Rec | F1 | AUC-ROC | AUC-PR |
|---|---|---|---|---|---|---|
| GIN | 63.34 ±1.48 | 63.23 ±0.81 | 63.74 ±5.51 | 63.38 ±2.71 | 68.02 ±0.74 | 59.91 ±0.77 |
| GAT | 71.45 ±2.24 | 73.58 ±0.79 | 66.84 ±5.58 | 69.96 ±3.38 | 81.30 ±0.31 | 78.68 ±0.34 |
| GCN | 69.39 ±0.79 | 65.89 ±1.78 | 80.96 ±5.03 | 72.51 ±1.31 | 76.48 ±0.61 | 74.41 ±2.34 |
| GraphSAGE | 70.17 ±4.13 | 67.20 ±5.52 | **81.56** ±4.91 | **73.32** ±1.80 | 77.10 ±0.78 | 71.09 ±2.87 |
| BrainIB | 52.48 ±5.32 | 36.64 ±24.44 | 60.17 ±45.02 | 43.77 ±31.08 | 62.14 ±2.65 | 54.56 ±3.19 |
| EEGNet | 62.03 ±0.18 | **77.13** ±1.34 | 62.03 ±0.18 | 55.91 ±0.39 | **87.10** ±4.52 | **88.60** ±3.82 |
| MedFormer | 54.70 ±10.26 | 56.93 ±12.01 | 48.35 ±20.99 | 49.78 ±13.95 | 57.37 ±15.34 | 58.27 ±11.66 |
| TopKPool | 68.71 ±6.64 | 67.98 ±4.73 | 70.29 ±19.41 | 67.75 ±11.67 | 71.64 ±9.07 | 63.99 ±5.71 |
| DiffPool | 62.56 ±7.74 | 67.84 ±10.98 | 51.47 ±22.75 | 55.08 ±16.11 | 70.81 ±10.10 | 68.82 ±9.90 |
| SAGPool | 58.90 ±7.91 | 57.74 ±5.92 | 58.51 ±18.26 | 57.50 ±11.89 | 63.93 ±0.19 | 54.13 ±0.07 |
| Cluster-GT | 68.12 ±0.49 | 65.94 ±0.50 | 75.00 ±2.47 | 70.15 ±0.94 | 72.66 ±0.45 | 62.68 ±0.62 |
| GraphPCA | 47.31 ±0.95 | 46.15 ±1.43 | 47.31 ±0.95 | 43.14 ±1.31 | 51.77 ±3.42 | 48.58 ±1.41 |
| ECoG-IBGT | **72.13** ±1.50 | 72.31 ±1.38 | 71.82 ±4.30 | 71.99 ±2.14 | 77.53 ±1.57 | 72.21 ±4.88 |

Table 3: Cross-subject AZ→BQ performance comparison (mean ± std, %). **Bold**: best; Underlined: second best.

**Ablation Study.** We evaluated six variants to quantify each component's contribution(see Table 4): the full model (ECoG-IBGT); a version without the mutual information estimation term $I(G_{\text{sub}}, G)$; one without the connectivity loss; and three ablations removing the node mask, edge mask, or subgraph generator. Each component substantially degrades performance when ablated, validating the rationale and effectiveness of the overall architecture.

**Parameter Selection.** We evaluated the sensitivity of our graph construction strategy to different functional connectivity measures and edge-retention thresholds. Results indicate that Pearson correlation offers the best balance between predictive accuracy and computational efficiency, with stable and competitive performance under a 10% edge-retention threshold (see Appendix I for details).

| Variant | Accuracy | Precision | Recall | F1-score | AUC-ROC | AUC-PR | Time (ms) |
|---|---|---|---|---|---|---|---|
| w/o MI estimation | 93.15 $\pm$1.43 | 98.79 $\pm$1.18 | 87.94 $\pm$3.65 | 92.99 $\pm$1.62 | 98.86 $\pm$0.27 | 99.08 $\pm$0.20 | 0.356 |
| w/o connectivity | 92.69 $\pm$1.31 | 98.71 $\pm$1.22 | 87.12 $\pm$3.45 | 92.50 $\pm$1.50 | 98.76 $\pm$0.30 | 98.98 $\pm$0.23 | 0.356 |
| w/o node masking | 80.62 $\pm$3.21 | **100.00 $\pm$0.00** | 62.69 $\pm$6.18 | 76.88 $\pm$4.89 | 98.37 $\pm$0.84 | 98.72 $\pm$0.75 | 0.355 |
| w/o edge masking | 84.61 $\pm$4.51 | 96.70 $\pm$5.12 | 73.81 $\pm$13.10 | 82.70 $\pm$6.34 | 97.35 $\pm$0.75 | 97.94 $\pm$0.47 | 0.354 |
| w/o subgraph gen. | 94.87 $\pm$1.12 | 98.25 $\pm$1.50 | 83.51 $\pm$2.08 | 94.11 $\pm$1.36 | 99.15 $\pm$0.36 | 99.12 $\pm$0.36 | **0.342** |
| **ECoG-IBGT** | **99.29 $\pm$0.30** | 99.88 $\pm$0.02 | **98.75 $\pm$0.07** | **99.31 $\pm$0.04** | **99.99 $\pm$0.01** | **99.99 $\pm$0.01** | 0.3623 |

Table 4: Ablation study results with standard deviations (%). **Bold**: best; Underlined: second best.

## 4.4 INTERPRETABILITY ANALYSIS (RQ3)

**Node Interpretability Analysis.** To investigate the contribution of individual channels to vocalization prediction, we visualized the average node mask (Fig. 5a). Node mask values were min–max normalized across all trials and channels to the $[0, 1]$ range, and a color bar indicates the mapping. Nodes with mask scores above 0.5 were rendered in red hues (greater importance), whereas those below 0.5 were shown in blue (lower importance). Results show vocalization in marmosets is associated with widespread cortical engagement. Within the auditory system, the rostral, caudal, and primary auditory cortices exhibited strong involvement. In the PFC, the dorsal prefrontal region emerged as a key node, consistent with prior reports (Kato et al., 2018). Notably, our masks consistently assigned very low weights to noisy electrodes, effectively suppressing their influence; details of the noise-channel identification and suppression process are provided in Appendix I.

**Edge Interpretability Analysis.** To distinguish the connectivity patterns between resting-state and vocalization, we compared the subgraph structures for each state. Specifically, we computed the mean edge importance across the 128-node functional graphs per state, retained the top 30 edges to form the dominant subgraphs, and visualized their differences (see subfigure (b) of Figure 5). Meanwhile, subfigure (a) presents normalized node-importance heatmaps for the PFC and A1 anatomical regions during each state. During vocalization, a circuit linking dorsolateral PFC (A46V) and primary somatosensory cortex (A3), with OPt as a multimodal hub (Andersen & Buneo, 2002; Colby & Goldberg, 1999), converges on auditory core and parabelt regions (AuA1, AuCPB) to form a predictive-control feedback loop. Frontal areas send corollary-discharge-like top-down signals to auditory cortex (Eliades & Wang, 2003; 2005; Tsunada et al., 2024), A3 provides somatosensory updates to motor plans (Tremblay et al., 2003), and auditory regions monitor vocal output for error correction (Tourville & Guenther, 2011). In rest, this loop is weakened and connectivity is instead dominated by a baseline fronto-auditory pathway between ventral PFC and rostral auditory fields, consistent with top-down modulation, predictive coding, and prior fronto-temporal findings (Friston, 2005; Asilador & Llano, 2021; Hockley et al., 2025a). Leveraging high-density, multi-region ECoG, we thus delineate fine-scale neural subcircuits supporting vocal production.

**Perturbation-based Validation.** To quantitatively verify that ECoG-IBGT truly relies on the subgraph motifs highlighted by the learned masks, rather than on diffuse, non-specific connectivity, we perform perturbation-based analyses on both nodes and edges (Figure 4). For node ablation, we progressively remove a small number of nodes (out of 128) according to two strategies: (i) `rmTop`, which drops nodes with the highest node-mask scores, and (ii) `rmRnd`, which removes the same number of nodes chosen uniformly at random. As shown in Figure 4a, classification accuracy decreases much more rapidly under `rmTop` than under `rmRnd`, while the unperturbed model (`base`) remains close to its original performance, indicating that the mask identifies functionally critical channels for anticipatory decoding.

For edge-level perturbations, we keep node features fixed and modify the graph structure under four settings: (i) `base`, the original graph; (ii) `rmTop`, removing edges with the highest edge-mask scores; (iii) `rmRnd`, randomly removing the same number of edges; and (iv) `addRnd`, randomly adding edges between previously unconnected node pairs. Figure 4b shows that deleting top-ranked edges (`rmTop`) leads to a pronounced and monotonic degradation in accuracy as the number of perturbed edges increases, whereas random deletions (`rmRnd`) or random edge additions (`addRnd`) induce substantially milder changes. The clear performance gap between structured and random

perturbations demonstrates that ECoG-IBGT depends on a compact set of behaviorally relevant subgraph motifs identified by the learned masks, providing subgraph-level interpretability for anticipatory decoding.

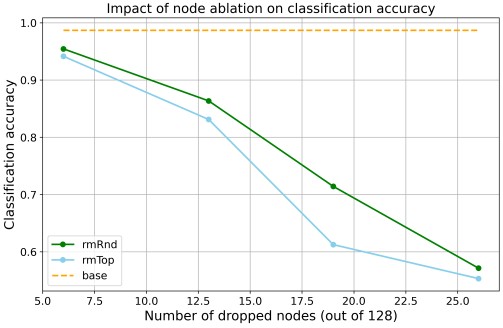 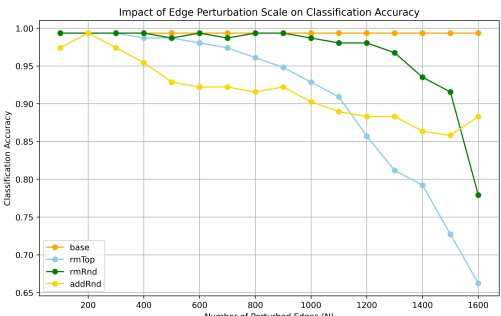

(a) Impact of node ablation on classification accuracy.

(b) Impact of edge ablation on classification accuracy.

Figure 4: Perturbation-based validation of node and edge masks. Accuracy drops much more steeply when removing top-ranked nodes/edges (`rmTop`) than under random perturbations (`rmRnd`, `addRnd`), indicating that the learned subgraph motifs are behaviorally critical.

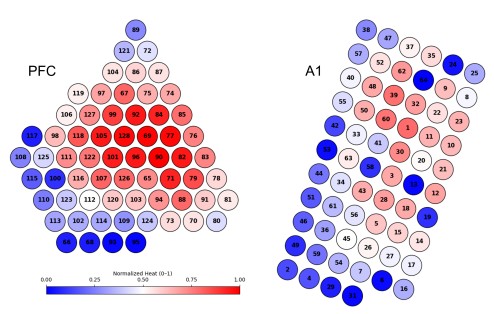 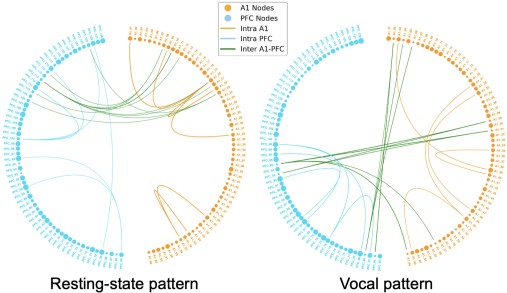

(a) Normalized node-importance heatmaps for PFC (left) and A1 (right) during rest and vocalization.

(b) Graph motifs of dominant connections in rest (left) and vocalization (right), intra-A1 (orange), intra-PFC (skyblue), and inter-A1–PFC (green).

Figure 5: (a) Node-mask heatmaps in PFC, A1; (b) Connection motifs for resting vs. vocal states.

## 5 CONCLUSION AND LIMITATIONS

In this work, we established a multi-dimensional experimental framework and collected high-quality implanted ECoG recordings from the primary auditory cortex (A1) and prefrontal cortex (PFC) in two adult common marmosets, encompassing pure-tone stimulation, conspecific call playback, and antiphonal vocal interactions. Based on these data, we developed an interpretable information-bottleneck graph transformer (ECoG-IBGT) for early decoding of high-density ECoG signals across multiple brain regions. Nonetheless, our findings are constrained by the inherent challenges of primate research. Common marmosets are costly to acquire and maintain, invasive ECoG systems require specialized development, and experimental scheduling must comply with strict ethical regulations. As a result, large-scale data collection comparable to rodent or non-invasive human studies remains impractical, and our dataset is limited in subject number and behavioral diversity. **Overall**, ECoG-IBGT achieved a peak accuracy of 99.29% at 400 ms prior to vocal onset, significantly surpassing 12 state-of-the-art baselines. More importantly, its interpretability enables the identification of task-relevant cortical subregions and distinct intra- and inter-regional connectivity motifs, thereby strengthening the neurophysiological foundation for anticipatory prediction. Future work will focus on expanding the cohort size, diversifying behavioral paradigms, and improving model adaptability to inter-individual variability, with the ultimate goal of achieving robust performance in naturalistic interactive settings.

ETHICS STATEMENT

We affirm compliance with the *ICLR Code of Ethics* and applicable institutional/regulatory policies governing research integrity, privacy, and human/animal subjects.

**Data, Privacy, and Consent.** All datasets used in this work are (i) publicly available under appropriate licenses, or (ii) collected with proper authorization. Where human or otherwise sensitive information may be present, we performed de-identification and adhered to relevant privacy regulations. In addition, all experimental procedures were approved by the Animal Use and Care Committee at [Institution Blinded for Review] and conducted in accordance with the National Institutes of Health Guidelines. The protocol number is blinded for review.

**Bias, Fairness, and Inclusivity.** We assessed potential sources of bias (e.g., class imbalance, domain shifts) and report group-wise performance where feasible. When disparities were observed, we document limitations and provide mitigation strategies (e.g., rebalancing, post-hoc calibration). The data licenses and collection processes are described to clarify provenance and potential sampling artifacts.

**Safety, Misuse, and Dual Use.** We discuss foreseeable risks of misuse (e.g., privacy attacks, misinformation, harmful automation) and outline safeguards (access control, audit logging, output filtering, human oversight) suitable for deployment contexts. We do not knowingly enable applications that violate laws, ethics, or user rights.

**Transparency and Conflicts of Interest.** We disclose funding sources, affiliations, and any relationships that could reasonably be perceived as conflicts of interest. All authors take responsibility for the integrity and accuracy of the results.

**Generative/Assistance Tools.** If large language models or similar tools were used for ideation, code drafting, writing, or editing, we include a separate "LLM Usage" note in the appendix specifying tools, versions, prompts/workflows at a high level, and human verification steps; all content remains the responsibility of the authors.

**Research Integrity.** We did not fabricate, falsify, or inappropriately manipulate data, results, or figures. All baselines are implemented or cited faithfully; deviations are documented. Any known limitations or negative findings are reported candidly.

*Ethics footprint.* We summarize ethical risks, mitigations, and residual concerns in the Limitations section to facilitate responsible follow-up work.

REPRODUCIBILITY STATEMENT

We have taken the following steps to facilitate reproduction and verification of our results:

**Code and Configuration.** Anonymized source code (training, inference, evaluation), exact hyperparameters, configuration files, and scripts are provided in the supplementary material and will be released upon publication.

**Data and Preprocessing.** We document data sources, licenses, filtering criteria, and splits. Exact preprocessing pipelines (feature extraction, normalization, windowing, augmentation) are described, with deterministic scripts to regenerate all artifacts. If redistribution is restricted, we provide retrieval instructions and hashing for integrity checks.

**Experimental Setup.** We report all key hyperparameters (optimizer, learning rate schedule, batch size, epochs, regularization), random seeds, model selection criteria, and early-stopping rules. Training/validation/test protocols and ablation settings are stated unambiguously.

**Multiple Runs and Uncertainty.** To account for stochasticity (initialization, sampling), we perform multiple runs with different seeds and report mean ± standard deviation for all primary metrics. Where feasible, we include calibration/error bars and confidence intervals.

**Hardware and Runtime.** We list hardware (e.g., GPU model and count, CPU, RAM), OS, and framework/library versions. We report training time, inference latency, and peak memory usage for the main models and key baselines.

**Theoretical Components (if any).** For claims involving theory, we provide complete assumptions, proofs, and derivations in the appendix, with pointers from the main text.

**Checklists and Artifacts.** We include a concise artifact table (code, configs, pretrained weights, logs, exact commands) and a README to reproduce each table/figure end-to-end. Any deviations from original baseline implementations are documented.

This statement references detailed instructions and pointers contained in the main paper, appendix, and supplemental materials, consistent with ICLR guidance that the reproducibility paragraph should *point to* the necessary details rather than duplicate them.

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

## A USE OF LLMS

In this work, our usage of large language models (LLMs) was strictly limited to language polishing and editorial assistance. No LLM was used in conceiving the research, designing the methodology, writing technical derivations, generating experiment code, or interpreting results.

Specifically:

- We used an LLM (e.g. GPT-4, or other) only to refine grammar, wording, phrasing, or clarity in certain prose sections.
- All substantive content—formulations, theorems, proofs, algorithmic design, experiments, data analysis—were authored and verified by the human authors.
- Any text generated by the LLM was carefully reviewed, edited, and fact-checked by the authors to ensure correctness and consistency with the intended meaning.
- Since the LLM usage played no role in the core scientific contributions, we do not treat it as a "contributor" under the ICLR LLM usage policy.

This approach ensures that all intellectual responsibility remains with the human authors, consistent with ICLR's policy requiring disclosure and accountability.

## B NOISE-CHANNEL DETECTION IN MARMOSET BQ

We apply four complementary diagnostics to verify that ECoG-IBGT's subgraph masks isolate noisy electrodes reliably.

**(a) Time-domain Variance (Figure 6a)** We compute each of the 128 electrodes' signal variance across the recording session. Light-gray bars represent all electrodes, and electrodes exceeding a variance Z-score of 1.5 (indicating abnormally high variance) are highlighted in red, marking them as potential noise channels. These elevated variance values typically correspond to artifacts such as electrical interference or poor electrode contacts.

**(b) Mean Inter-Channel Correlation (Figure 6b)** Next, we calculate the mean Pearson correlation coefficient of each electrode's signals with all other channels. Electrodes exhibiting mean correlations below 0.5 (marked in red) suggest significantly reduced signal consistency and synchrony with neighboring electrodes. Importantly, many of these low-correlation electrodes overlap with those identified by variance, reinforcing the identification of these channels as genuine noise sources.

**(c) Power Spectral Density(PSD) Comparison (Figure 6c)** To further validate channel noise, we examine the PSD of signals from flagged channels (red) and compare them with known clean electrodes (blue). Although both channel types show characteristic power-line interference at harmonics of 50 Hz (50, 100, and 150 Hz), noise channels exhibit markedly higher amplitude peaks and a pronounced elevation across a broad frequency spectrum. This broadband increase in noise floor indicates genuine noise contamination rather than typical physiological variations.

**(d) Spatial Z-score Mapping (Figure 6d)**  Mapping the variance Z-scores spatially onto the $8,\times,16$ electrode grid allows visualization of the noise distribution. Notably, high-noise electrodes (dark red) appear spatially clustered rather than randomly scattered, suggesting a localized hardware or electrode contact problem. This pattern is critical for interpreting noise sources, as spatial clustering typically implicates technical rather than biological factors.

Centering the thresholding criterion, we define:

$$\text{noise\_idx} = \big\{\, i \mid |z_{\text{var},i}| > 1.5 \ \lor \ \text{mean\_corr}_i < 0.5 \big\}, \tag{12}$$

where $z_{\text{var},i}$ is the variance Z-score and $\text{mean\_corr}_i$ is the mean Pearson correlation of channel $i$.

Figure 7 repeats these diagnostics for Marmoset AZ with identical thresholds and layouts. Direct comparison between Figures 6 and 7 confirms the reliability of our approach. Both subjects exhibit consistent identification of severely affected electrodes, evident in similarly exaggerated PSD profiles.

To summarize clearly and succinctly, Table 5 lists all identified noisy electrode indices for both subjects. This structured detection and validation process ensures the accuracy of downstream neural analyses by eliminating channels compromised by extraneous noise.

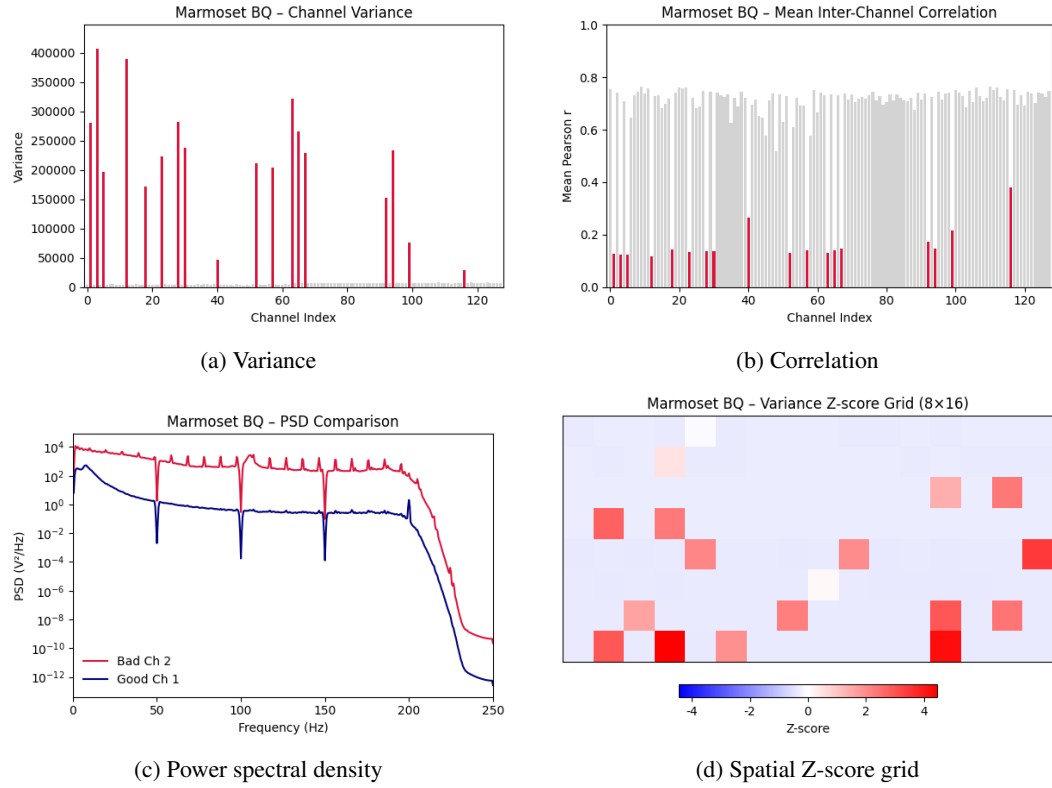

(a) Variance

(b) Correlation

(c) Power spectral density

(d) Spatial Z-score grid

Figure 6: Noise-channel detection in Marmoset BQ: (a) variance, (b) mean inter-channel correlation, (c) power spectral density comparison, (d) spatial Z-score grid.

Table 5: Detected noisy channels for Marmoset BQ and AZ (1-based indices).

| BQ | AZ |
| --- | --- |
| 2, 4, 6, 13, 19, 24, 29, 31, 41, 53, 58, 64, 66, 68, 93, 95, 100, 117 | 2, 4, 7, 29, 31, 41, 49, 66, 68, 74, 85, 87, 93, 95 |

These comprehensive diagnostics (variance, correlation, PSD, and spatial analyses) collectively demonstrate our pipeline's efficacy in identifying, validating, and localizing noisy electrodes, thus

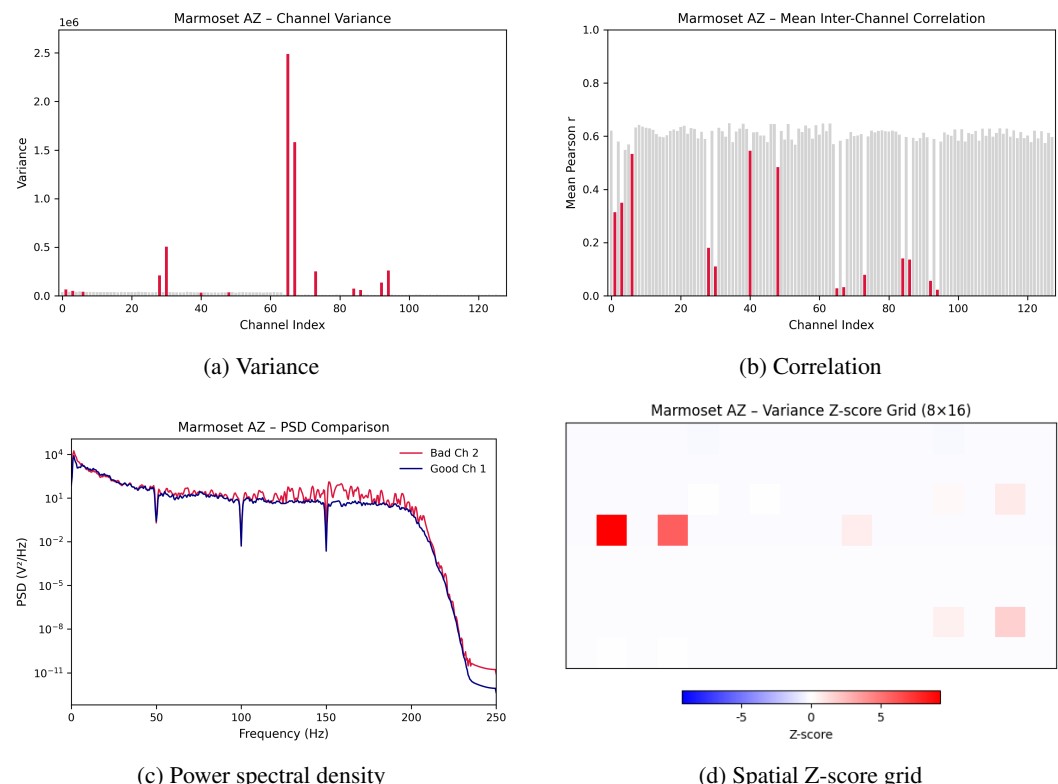

(a) Variance

(b) Correlation

(c) Power spectral density

(d) Spatial Z-score grid

Figure 7: Noise-channel detection in Marmoset AZ: same diagnostics and thresholds as in Figure 6.

ensuring that the subsequent analyses are focused exclusively on physiologically meaningful ECoG signals.

## C    VISUALIZATION OF NOISE-CHANNEL SUPPRESSION

This appendix provides a detailed visualization illustrating how ECoG-IBGT effectively suppresses activations in channels previously identified as noisy. Figures 8a and 8b present normalized latent activation heatmaps for subjects Marmoset BQ (a) and AZ (b), respectively. Activation values are scaled between 0 and 1 to facilitate consistent comparison across all channels.

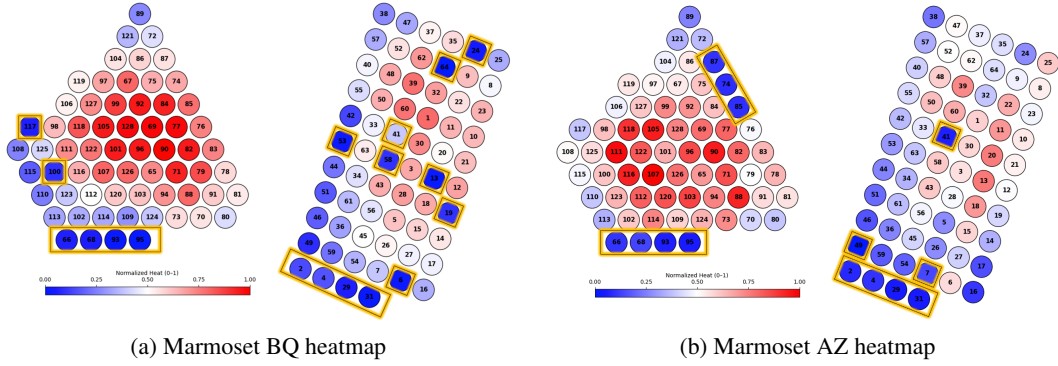

(a) Marmoset BQ heatmap

(b) Marmoset AZ heatmap

Figure 8: Latent activation heatmaps showing suppression of noise channels. Activations normalized to [0,1]; warmer colors indicate stronger activation. Previously flagged noisy channels are outlined in yellow.

**Identification of Noise Channels**  Channels previously marked as noisy (refer to Table 5 in Appendix A) are highlighted with yellow outlines in each heatmap. This clear demarcation enables direct visual assessment of the model's suppression effectiveness relative to previously established noise locations.

**Effectiveness of Suppression.**  Across both animals, channels identified as noisy consistently display minimal activation levels (represented by cooler colors, primarily blue). In contrast, adjacent physiological channels exhibit broader and higher activation ranges (green to red), indicating intact and meaningful signal representation. This clear distinction quantitatively demonstrates that the masking mechanism of ECoG-IBGT effectively attenuates noise-contaminated channels without adversely impacting the integrity of physiologically relevant signals.

**Robustness Across Trials.**  ensure reliability and mitigate variability inherent to single-trial data, we averaged latent activations over 50 randomly selected trials per subject. The resulting averaged heatmaps reveal a consistent pattern of suppression across multiple stimuli and recording sessions, reinforcing the robustness and generalizability of our approach.

**Implications for Analytical Interpretability**  By substantially reducing the influence of noise channels, our method significantly enhances the reliability and interpretability of downstream graph-level analyses. Consequently, this targeted suppression not only optimizes predictive accuracy but also ensures that physiological interpretations derived from subgraph analyses accurately reflect true neural dynamics, untainted by artifacts.

## D  HYPERPARAMETER SENSITIVITY ANALYSIS

We evaluate sensitivity to key hyperparameters. Figures 9, 10, and 11 show their effects on core metrics.

**Information Bottleneck and Connectivity Weights.**  Figure 9 shows optimal settings for MI weight $\alpha$ and connectivity loss weight $\beta$ at $10^{-5}$; deviations degrade all metrics.

**Gumbel–Softmax Mask Temperatures.**  Figure 10 indicates both edge and node temperature optimals at 0.5; improper temperatures reduce subgraph quality.

**Pooling Readout.**  We performed comprehensive sensitivity analyses to evaluate the robustness of ECoG-IBGT concerning critical hyperparameters. Figures 9, 10, and 11 illustrate the effect of varying hyperparameters on evaluation metrics including Accuracy, Precision, Recall, F1-score, AUC-ROC, and AUC-PR.

**Information Bottleneck and Connectivity Loss Weights.**  As shown in Figure 9, both the mutual information (MI) term weight ($\alpha$) and connectivity loss weight ($\beta$) have optimal settings at $10^{-5}$. Deviating from this optimum negatively impacts all metrics, indicating the importance of balanced information capture and subgraph connectivity.

**Gumbel–Softmax Mask Temperatures.**  Figure 10 indicates optimal edge-mask and node-mask temperatures are both at 0.5. Lower or higher values result in gradual performance deterioration, demonstrating that proper mask temperature is essential for maintaining predictive subgraph quality and mask sparsity.

**Pooling Readout Functions.**  As illustrated in Figure 11, max pooling consistently outperforms mean and sum pooling across all metrics. This suggests that emphasizing the strongest edge contributions yields the most discriminative and effective subgraph representations.

## E  NOTATION AND ALGORITHMIC DETAILS

We use the following notation: $N$ is the number of nodes; $E$ the number of edges; $F$ the per-node feature dimension; $L$ TransformerConv layers; $M$ MLP classifier layers (including output);

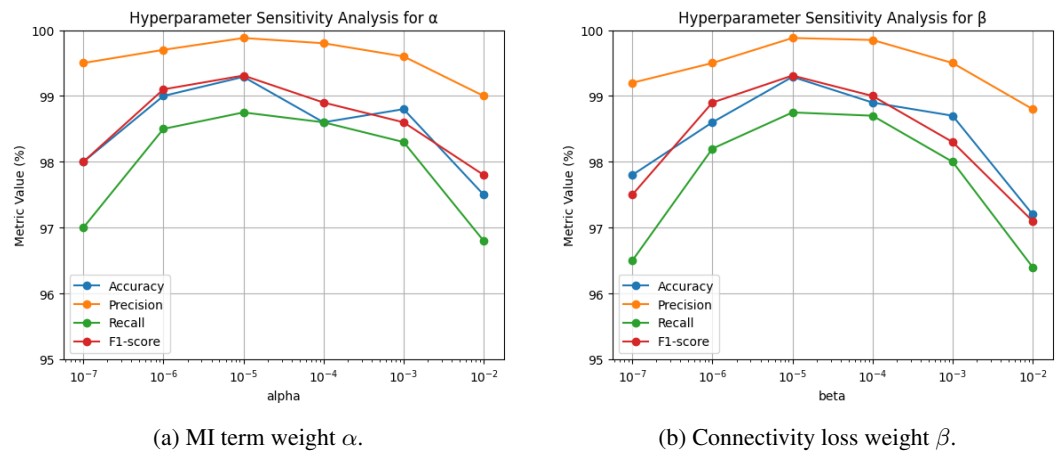

(a) MI term weight $\alpha$.

(b) Connectivity loss weight $\beta$.

Figure 9: Sensitivity of information bottleneck and connectivity loss weights.

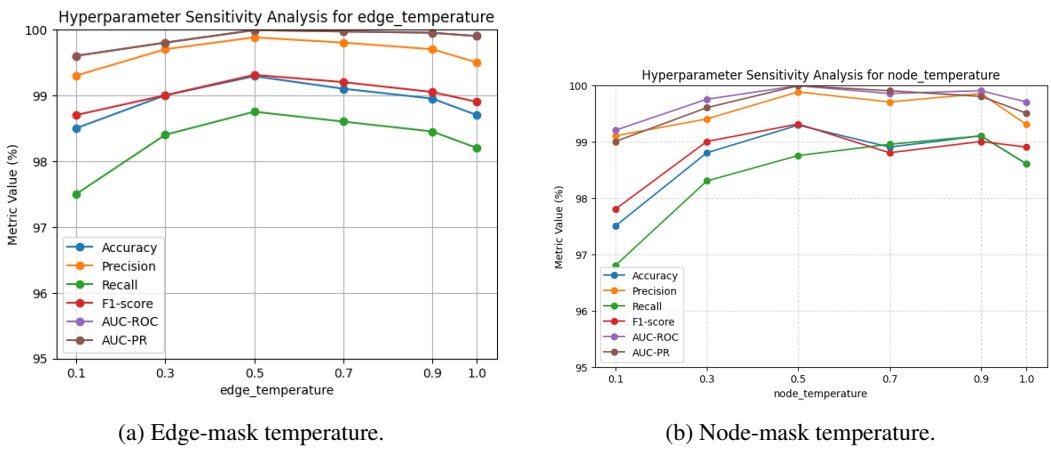

(a) Edge-mask temperature.

(b) Node-mask temperature.

Figure 10: Sensitivity of Gumbel–Softmax temperatures for masks.

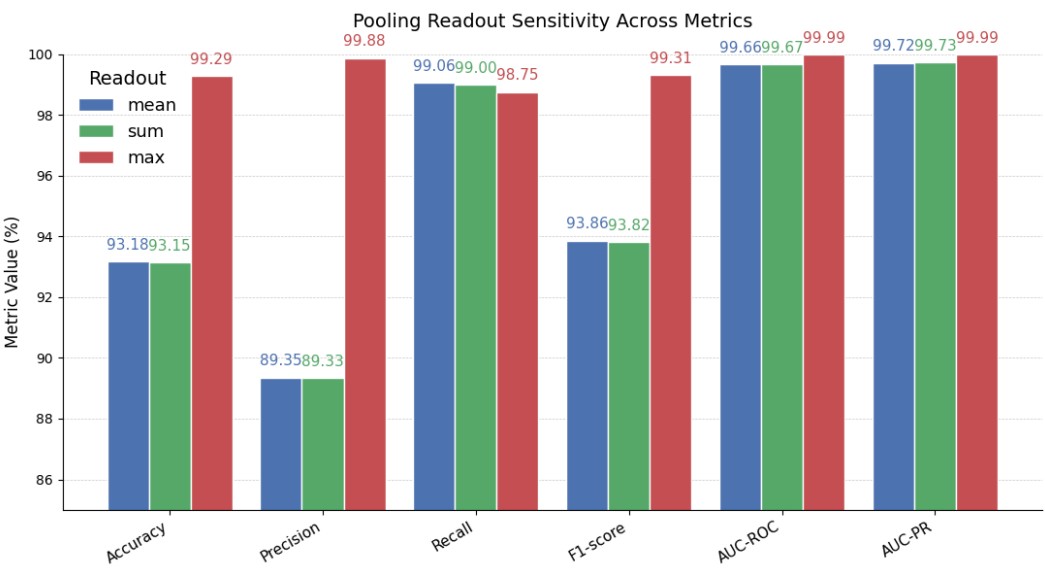

Figure 11: Comparison of pooling readout functions. Max pooling yields best performance across metrics.

$H$ attention heads per layer; $D$ the final node embedding dimension (`latent_dim[-1]` $\times H$); $B$ batch size; $C$ output classes; and $\tau_e, \tau_n$ the Gumbel–Softmax temperatures for edge- and node-mask sampling.

Below we give pseudocode and concise explanations for the two core components: the IB–Subgraph Generator (Algorithm 1) and the Dual Structure Graph Encoder (Algorithm 2).

**IB–Subgraph Generator (Algorithm 1).** This module generates differentiable masks to highlight important nodes and edges for subgraph extraction. It operates by first concatenating node feature pairs for each edge to form edge-specific features, which are then passed through an edge-level multilayer perceptron (EdgeMLP). Node features independently go through a node-level MLP (NodeMLP). During training, Gumbel-Softmax sampling ensures differentiability by introducing stochastic noise controlled by temperatures $\tau_e$ and $\tau_n$, enabling gradient-based optimization of discrete masks. In inference, deterministic sigmoid activation is applied for mask generation.

---

**Algorithm 1** IB–Subgraph Generator

---

**Input:** Node features $X \in \mathbb{R}^{N \times F}$, edge indices `edge_index` $\in \mathbb{Z}^{2 \times E}$, edge temperature $\tau_e$, node temperature $\tau_n$, training flag
**Output:** Edge mask $m_e \in [0,1]^E$, node mask $m_n \in [0,1]^N$
  1: Transfer inputs to device
  2: **for** each edge $k = 1, \ldots, E$ **do**
  3:     $(i,j) \leftarrow$ `edge_index`$[:, k]$
  4:     *expl_in*$[k] \leftarrow [X[i]; X[j]]$
  5: **end for**
  6: $w_e \leftarrow \text{EdgeMLP}(\textit{expl\_in})$
  7: $w_n \leftarrow \text{NodeMLP}(X)$
  8: **function** SAMPLE$(w, \tau)$
  9:     **if** training **then**
 10:         Sample Gumbel noise $\varepsilon \sim \text{Uniform}(10^{-5}, 1 - 10^{-5})$
 11:         $g \leftarrow \ln(\varepsilon) - \ln(1 - \varepsilon)$
 12:         $\ell \leftarrow (w + g)/\tau$
 13:         **return** $\sigma(\ell)$
 14:     **else**
 15:         **return** $\sigma(w)$
 16:     **end if**
 17: **end function**
 18: $m_e \leftarrow \text{Sample}(w_e, \tau_e)$, $m_n \leftarrow \text{Sample}(w_n, \tau_n)$
 19: **return** $m_e, m_n$

---

**Dual Structure Graph Encoder (Algorithm 2).** This encoder leverages Transformer-based convolutional layers (TransformerConv) to learn node embeddings. The learned embeddings undergo normalization (BatchNorm) and activation (ReLU) before applying masks generated by the IB–Subgraph Generator. Graph-level embeddings are computed via global max pooling. An MLP classifier processes these embeddings, consisting of multiple linear transformations interleaved with batch normalization, ELU activations, and dropout regularization, culminating in the final prediction logits.

## F    PERTURBATION EXPERIMENTS FOR INTERPRETABILITY

To further evaluate the interpretability of our model, we conducted perturbation experiments by systematically altering the graph structure and observing the corresponding changes in classification accuracy. Specifically, we examined four scenarios:

- **base**: The original unperturbed graph.
- **rmTop**: Removing the most important edges ranked by the learned edge mask.
- **rmRnd**: Randomly removing the same number of edges as in **rmTop**.

---

**Algorithm 2** Dual Structure Graph Encoder

---

**Input:** Node features $X$, edge index $E$, batch indices $B$, edge weights $W_e$ (optional), node mask $m_n$ (optional)
**Output:** Logits, graph embedding $z_g$, node embeddings $Z_n$
 1: Initialize missing $W_e, m_n$ to ones
 2: **for** layer $i = 1$ to $L$ **do**
 3:     $X \leftarrow \text{TransformerConv}_i(X, E, W_e)$
 4:     $X \leftarrow \text{BatchNorm}_i(X)$
 5:     $X \leftarrow \text{ReLU}(X)$
 6: **end for**
 7: $Z_n \leftarrow m_n \odot X$
 8: $z_g \leftarrow \text{global\_max\_pool}(Z_n, B)$
 9: $h \leftarrow z_g$
10: **for** MLP layer $i = 1$ to $M - 1$ **do**
11:     $h \leftarrow \text{Linear}_i(h)$
12:     $h \leftarrow \text{BatchNorm}_i(h)$
13:     $h \leftarrow \text{ELU}(h)$
14:     $h \leftarrow \text{Dropout}(h)$
15: **end for**
16: $\text{logits} \leftarrow \text{Linear}_M(h)$
17: **return** $\text{logits}, z_g, Z_n$

---

• **addRnd**: Randomly adding new edges between unconnected node pairs.

As shown in Figure 12, the model demonstrates a clear robustness pattern. When important edges (**rmTop**) are removed, classification accuracy degrades rapidly, confirming that the identified edges are indeed critical for predictive performance. In contrast, randomly removing edges (**rmRnd**) has a more gradual effect, while random edge additions (**addRnd**) lead to moderate performance declines, especially at larger perturbation scales. Notably, the **base** model maintains consistently high accuracy, serving as the upper bound.

These results indicate that the subgraph masks produced by our model successfully capture structurally meaningful connections: perturbing edges deemed important by the model has a much stronger impact than perturbing random edges. This validates the interpretability of the learned subgraphs and further highlights the model's ability to reveal functionally relevant neural circuit motifs.

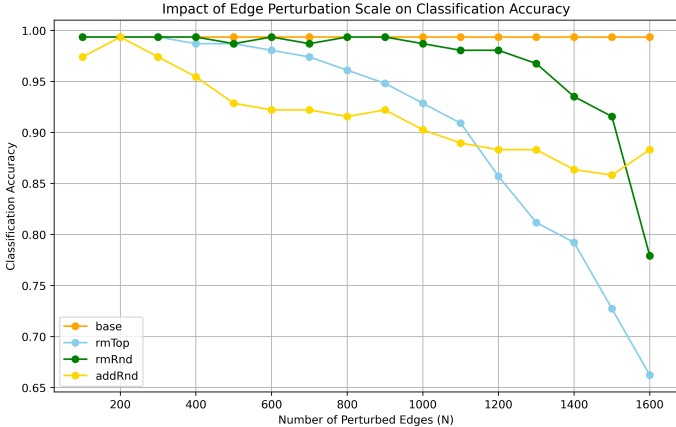

Figure 12: Impact of edge perturbation scale on classification accuracy. **base**: no perturbation; **rmTop**: removing important edges identified by the model; **rmRnd**: randomly removing the same number of edges; **addRnd**: randomly adding edges between previously unconnected nodes.

## G  ADDITIONAL BENCHMARK: BCI COMPETITION III DATASET I

**Protocol.** To further evaluate generalization beyond our in-house marmoset dataset, we test ECoG-IBGT on the public **BCI Competition III, Dataset I**Lal et al. (2004). This dataset records imagined finger versus tongue movements from invasive ECoG grids (278 training trials, 100 test trials, 64 channels, 1000 Hz sampling). Unlike our main experiments, we adopt a *minimalist* configuration: (1) functional connectivity computed by Pearson correlation, (2) binary adjacency via top-50% thresholding, (3) no additional "strong" features (e.g., high-gamma power, LMP, PSD), (4) no hyperparameter tuning (default: batch size 128, epochs 300, learning rate $5 \times 10^{-5}$).

| Split | Acc | Prec | Rec | F1 | AUC-ROC | AUC-PR |
|---|---|---|---|---|---|---|
| Val | $0.7143 \pm 0.0000$ | $0.8776 \pm 0.0365$ | $0.5467 \pm 0.0452$ | $0.6713 \pm 0.0153$ | $0.7638 \pm 0.0102$ | $0.7988 \pm 0.0117$ |
| Test | $0.7857 \pm 0.0452$ | $0.7339 \pm 0.0588$ | $0.7273 \pm 0.0643$ | $0.7276 \pm 0.0466$ | $0.8219 \pm 0.0202$ | $0.7531 \pm 0.0187$ |

Table 6: ECoG-IBGT on BCI Competition III Dataset I without strong features or parameter tuning.

**Observations.** Even under this simplified setup, our model achieves **78.6% test accuracy** with **AUC-ROC 0.82** and balanced precision/recall. This demonstrates that ECoG-IBGT can extract discriminative subgraphs without task-specific feature engineering or dataset-specific tuning. The results confirm the *positive transferability* of our framework across benchmarks.

## H  CROSS-SESSION EVALUATION

To further assess generalization across distinct experimental sessions, we conducted **cross-session decoding experiments** using different trial groups as training and test sets. Specifically:

- **Trial-3 → Trial-4**: model trained on Trial-3 (antiphonal calling, 1012 trials) and tested on Trial-4 (mixed playback sessions, 1012 trials);

- **Trial-4 → Trial-3**: reciprocal configuration with Trial-4 as training and Trial-3 as testing;

- **Trial-5 → Trial-6**: model trained on Trial-5 (free-behavior session, 222 trials) and tested on Trial-6 (chair-restrained session, 222 trials);

- **Trial-6 → Trial-5**: reciprocal configuration with Trial-6 as training and Trial-5 as testing.

Table 7 summarizes the cross-session validation and testing accuracy. We note that, despite the distributional shifts across recording conditions (antiphonal calling vs. pure-tone/conspecific playback; free-behavior vs. chair-restrained), our method maintains robust performance, achieving up to **95.98%** cross-session accuracy.

| Training → Testing | Validation Acc. | Cross-Session Acc. |
|---|---|---|
| Trial-3 → Trial-4 | 98.91% | 80.08% |
| Trial-4 → Trial-3 | 98.37% | 61.83% |
| Trial-5 → Trial-6 | 100.00% | 95.98% |
| Trial-6 → Trial-5 | 100.00% | 81.10% |

Table 7: Cross-session evaluation results on different trial groups.

## I  ADDITIONAL ANALYSIS ON GRAPH CONSTRUCTION SENSITIVITY

Our graph construction strategy is based on Pearson correlation with a fixed top-$k$ threshold (10%). We note that this choice—of threshold, the use of absolute values, and the length of the time window—may affect downstream performance. To assess robustness, we performed two complementary experiments.

**(1) Comparing different functional connectivity measures.** We evaluated whether the choice of connectivity metric substantially affects performance or computational efficiency. We compared Pearson correlation, coherence, phase-locking-value, and power spectral density (PSD) similarity, each retaining the top 30% of edges. Results are summarized in Figure 13. While accuracy is broadly similar (94%–98.5% across methods), runtime per sample differs significantly. Pearson correlation achieves both strong accuracy (96.65%) and the fastest efficiency (34.23 ms per sample), whereas coherence and PSD similarity provide slightly higher accuracy (97.2%–97.5%) at the cost of nearly an order of magnitude higher computation time. Based on this analysis, we adopt Pearson correlation for subsequent experiments.

**(2) Varying edge-retention thresholds.** We next assessed how performance changes when varying the proportion of retained edges under the Pearson correlation metric. Specifically, we varied the remaining percentage of top-$k$ edges from 5% to 30% while keeping other parameters fixed (absolute value, window size). As shown in Figure 13, classification performance remains highly stable across thresholds. Both AUC-ROC and AUC-PR consistently remain above 0.997, while accuracy fluctuates only within ±0.01. This suggests our model is robust to the specific choice of edge-retention threshold.

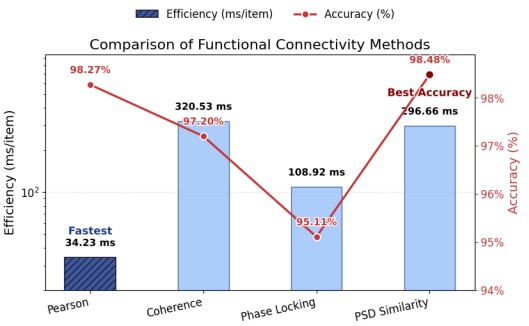

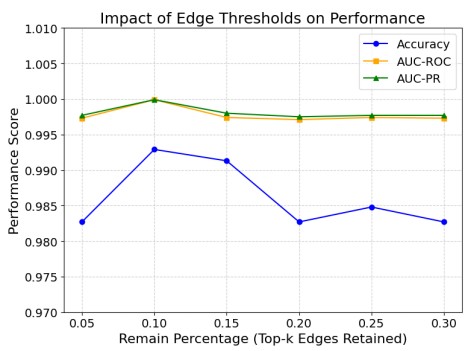

(a) Comparison of functional-connectivity measures (top 30% edges retained).

(b) Effect of varying edge-retention thresholds under Pearson correlation.

Figure 13: (a) Performance vs. efficiency for various connectivity measures. (b) Sensitivity of classification performance to different edge-retention thresholds under Pearson correlation.

**Summary.** Together, these analyses show that (i) while alternative connectivity measures may yield marginal accuracy gains, Pearson correlation provides the best trade-off between predictive accuracy and computational efficiency; and (ii) under this metric, model performance is stable across a range of edge-retention thresholds. These findings strengthen our conclusions about the robustness of graph-construction choices.

