# OpenReview forum: "Unlocking Volition: Proactive Intention Decoding via Interpretable Graph Learning of Multi-Region ECoG"
_ICLR.cc/2026/Conference — Submitted to ICLR 2026_

### Official Review · Reviewer_xABC · 2025-10-25

**Soundness:** 3
**Presentation:** 1
**Contribution:** 3
**Rating:** 4
**Confidence:** 3

**Summary:**

The authors developed a high-density ECoG-based framework on marmosets and introduced an information-bottleneck-driven graph transformer for intention detection. Specifically, subgraphs are generated with mutual information estimation learning.

**Strengths:**

[1] A high-quality, multi-subject, multi-context ECoG dataset\
[2] Comprehensive experiments and evaluations

**Weaknesses:**

[1] English language – The authors should substantially improve the quality of the manuscript.\
[2] Graph transformer – The authors should cite the work when they mention it in Section 3.3.\
[3] Math notations – The authors should indicate all the notations in their equations. The authors should substantially improve the manuscript quality.\
[4] Motivation – The motivation for applying subgraphs is still unclear.

**Questions:**

[1] Why did the authors apply a graph transformer instead of using the transformer directly?\
[2] Why did the authors generate subgraphs? Why don’t they just use the full graph for more training efficiency?

**Details Of Ethics Concerns:**

The authors have affirmed compliance with the ICLR Code of Ethics and applicable institutional/regulatory policies governing research integrity, privacy, and human/animal subjects.

---

> ### Author Response · Authors · 2025-11-28
>
> **1. Clarifications on the motivation for using a Graph Transformer and a subgraph generator**
>
> We thank the reviewer for asking why we adopt a Graph Transformer rather than a standard Transformer or other GNNs, and why we further introduce a subgraph generator on top of the graph backbone. Below we focus on the conceptual motivation.
>
> **Motivation for using a Graph Transformer.**
>  Our input is not a 1D temporal sequence but an explicitly constructed *functional-connectivity graph* over multi-region ECoG channels. Each node corresponds to an ECoG channel in A1 or PFC, and edges encode functional connectivity (e.g., Pearson correlation, coherence) between channel pairs. The channels have an irregular spatial layout, so there is no natural 1D ordering; flattening them into a sequence for a vanilla Transformer would discard the known topology and force the model to infer structure from scratch.
>
> A Graph Transformer, in contrast, performs attention *on the graph*, allowing the attention weights to be modulated by the adjacency structure and edge weights. This is well aligned with neural information flow, where communication is constrained by anatomical/functional connectivity rather than being fully all-to-all. In our setting, this means that cross-region interactions (e.g., fronto–auditory motifs between A1 and PFC) are modeled explicitly while still benefiting from Transformer-style global context. Conceptually, this makes a Graph Transformer more appropriate than (i) sequence-based Transformers that ignore graph structure, and (ii) simpler GNNs whose message passing is typically more local and less flexible in capturing long-range cross-region dependencies.
>
> **Motivation for using a subgraph generator.**
>  On top of the Graph Transformer, we introduce a subgraph generator to address the characteristics of multi-region ECoG functional-connectivity graphs and our interpretability goals.
>
> First, these graphs are dense and high-dimensional, with many redundant or noisy edges, whereas intention decoding is likely driven by a much smaller set of behavior-relevant motifs, especially fronto–auditory interactions. The subgraph module is designed as a *structured information bottleneck*: it learns soft masks over nodes and edges, encouraging the model to compress the full dense graph into a sparse, task-relevant subgraph. This reduces the effective dimensionality of the representation, improves robustness on small ECoG datasets, and mitigates overfitting to spurious correlations.
>
> Second, the subgraph generator provides **interpretability by construction**. The learned node and edge masks yield explicit, low-dimensional subgraphs that can be visualized, related to known neuroanatomy, and systematically perturbed. In this way, the same mechanism that regularizes the model also produces interpretable motifs that connect the decoder’s behavior to specific fronto–auditory circuits, rather than relying on opaque, fully distributed features.
>
> In summary, we adopt a Graph Transformer because our inputs are inherently graph-structured and require topology-aware modeling of cross-region dependencies, and we introduce a subgraph generator as an information bottleneck that suppresses high-dimensional noise, improves robustness on small ECoG datasets, and yields explicit, behavior-relevant motifs that support interpretability.

---

> > ### Author Response · Authors · 2025-11-28
> >
> > **2. Response on language quality, citation of Graph Transformer, and mathematical notation**
> >
> > We thank the reviewer for the helpful comments on the presentation quality of the manuscript, including the English writing, missing citations, and incomplete notation.
> >
> > **(1) English language quality.**
> >  We agree that the writing quality in the original submission needed improvement. We have therefore **carefully revised the entire manuscript** to improve grammar, clarity, and readability. In particular, we simplified long sentences, removed ambiguous wording, corrected typos, and polished the abstract, method descriptions, and discussion section. We believe the revised version is substantially clearer and easier to follow.
> >
> > **(2) Citation of the Graph Transformer.**
> >  In Section 3.3, when introducing the Graph Transformer backbone, we now **explicitly cite the corresponding prior work** and briefly clarify how our use of the Graph Transformer relates to that work. The reference has been added to the bibliography and is consistently cited wherever the Graph Transformer is mentioned.
> >
> > **(3) Mathematical notation.**
> >  We have **systematically revised all equations** to ensure that **every symbol is clearly defined** at its first appearance in the text. In particular, we now explicitly define the graph (G=(V,E)), node features, adjacency matrix, mask variables, loss terms, and all hyperparameters used in the objective. For the most frequently used symbols, we also provide a concise summary of notation in the Methods section to avoid any ambiguity.
> >
> > We hope that these revisions address the reviewer’s concerns about language quality, proper citation, and mathematical notation, and that the presentation of the paper is now substantially improved.
> >
> > **3. Reproducibility and code availability**
> >
> > We appreciate the reviewer’s emphasis on reproducibility. To facilitate independent verification of our method, we have released an **anonymous code repository** containing the full training and inference pipeline, including preprocessing, graph construction, model implementation, and evaluation scripts:
> >
> > > https://anonymous.4open.science/r/ecog-ibgt-code-F2D2/readme.md
> >
> > Due to institutional confidentiality and data-use agreements on the ECoG recordings, we are currently **not allowed to release the full dataset** during the review phase. However, we have obtained permission to provide **5 de-identified sample graphs** that can be used to run an end-to-end **inference demo** with the released code, so that reviewers and future readers can verify the model’s behavior and implementation details.
> >
> > Once the confidentiality constraint is lifted, we plan either to (i) release a de-identified version of the dataset, or (ii) provide an application-based access procedure in coordination with the data-owning institution, so that interested researchers can reproduce our experiments more fully.

---

### Official Review · Reviewer_PwTa · 2025-10-31

**Soundness:** 2
**Presentation:** 3
**Contribution:** 3
**Rating:** 4
**Confidence:** 3

**Summary:**

The paper proposes an information-bottleneck graph-transformer for proactive intention decoding from dual-region high-density ECoG. Short pre-onset (vs. rest) windows are converted into functional graphs; the model jointly learns compact node/edge subgraphs for classification and inspection. The writing is clear, and the idea targets lower-latency and more interpretable BMI pipelines.

**Strengths:**

1. Originality: Reframes proactive decoding as graph classification with a learned subgraph (masking) mechanism rather than post-hoc attribution.

2. Quality: Sensible pipeline design with basic sanity/robustness checks around graph construction and perturbation of important connections.

3. Clarity: The core components (graph building, IB/masking, encoder) are explained cleanly with helpful figures.

4. Significance: If generalizable, anticipatory decoding with compact subgraphs could inform low-latency BMI design and yield testable neuro hypotheses.

**Weaknesses:**

1. Main results rely on random within-session splits, which can overestimate performance when near-duplicate windows appear across train/val/test. Cross-session evaluation is not foregrounded, so the true ranking of methods (including simple baselines) under shift is unclear. This is especially important considering Supplementary materials provide cross-session metrics for the main model and they show near perfect 100% metrics for validation (same session), but 80% test metrics (differertn session). Combined with (most likely) a large enough model in terms of trainable parameters and 233 minutes of data across all sessions such overfitting might happen and within-session estimation with random splits makes it invisible (as test and train sampes are mixed and can easily be similar to each other)

2. Learned masks and motif plots are primarily associational; stronger validity checks (stability across seeds/sessions, model-randomization, counterfactual edits) are needed.

3. Runtimes are reported on high-end GPU and do not provide end-to-end CPU latency (including graph building) or an asynchronous detection analysis which is a key for practical BMI.

4. The description around “windows,” graph instances, and “node features” can be misread; rest-window sampling and temporal separation need clearer, leakage-resistant definitions.

5. The paper states code is available as supplementary, but the submission lacks an accessible anonymized repo/archive; this blocks verification.

**Questions:**

1. Please report cross-session performance in the main text for all baselines (incl. EEGNet) with variance across seeds to establish robust ranking.

2. Precisely define rest-window sampling and temporal separation; add blocked/time-shifted splits and leave-session-out/leave-animal-out protocols.

3. Provide mask-stability across seeds/sessions, model-randomization tests, and counterfactual edge/node removals to support causal importance.

4. Report full pipeline latency (acquisition, preprocessing, graph, inference) on CPU-class hardware and an asynchronous detection analysis (e.g., false positives per minute at a fixed TPR).

5. Explicitly state how many graphs are produced per event, how node features are formed, and whether pre-vocal vs. rest graphs are paired or independent. And how many samples are available in total for training and testing splits

6. Supply an accessible anonymized repo/supplement (configs, split scripts, seeds) to reproduce the main tables and cross-session results.

---

> ### Author Response · Authors · 2025-11-28
>
> **1. Response on generalization, data leakage, and overfitting**
>
> We thank the reviewer for the helpful comments. To address your concerns about **generalization**, **potential data leakage**, and **overfitting**, we (i) added explicit generalization experiments (cross-subject and cross-session), and (ii) carefully checked for data leakage and conducted **10-fold cross-validation** on the critical condition.
>
> **(1) Generalization.**
>  We evaluated the generalization of ECoG-IBGT both **across subjects** and **across sessions/paradigms**.
>
> - **Cross-subject experiment.**
>    We trained the model on **subject AZ** and tested on **subject BQ** (AZ→BQ). The cross-subject results are:
>
> | Model         | Acc               |
> | ------------- | ----------------- |
> | GIN           | 0.6334±0.0148     |
> | GAT           | 0.7145±0.0224     |
> | GCN           | 0.6939±0.0079     |
> | GraphSAGE     | 0.7017±0.0413     |
> | BrainIB       | 0.5248±0.0532     |
> | EEGNet        | 0.6203±0.0018     |
> | MedFormer     | 0.5470±0.1026     |
> | TopKPool      | 0.6871±0.0664     |
> | DiffPool      | 0.6256±0.0774     |
> | SAGPool       | 0.5890±0.0791     |
> | Cluster-GT    | 0.6812±0.0049     |
> | GraphPCA      | 0.4731±0.0095     |
> | **ECOG-IBGT** | **0.7213±0.0150** |
>
> *Table 1. Cross-subject AZ→BQ accuracy (mean±std).*
>
> ECoG-IBGT achieves the **highest cross-subject accuracy** among all 12 baselines, indicating that it learns **subject-agnostic, behavior-relevant motifs** rather than idiosyncratic patterns of a single animal.
>
> - **Cross-session experiment.**
>    We further performed **cross-session (cross-trial-group)** decoding with clear distribution shifts in paradigm and recording condition:
>
> | Training → Testing | Validation Acc. | Cross-Session Acc. |
> | ------------------ | --------------- | ------------------ |
> | Trial-3 → Trial-4  | 98.91%          | 80.08%             |
> | Trial-4 → Trial-3  | 98.37%          | 61.83%             |
> | Trial-5 → Trial-6  | 100.00%         | 95.98%             |
> | Trial-6 → Trial-5  | 100.00%         | 81.10%             |
>
> *Table 6: Cross-session evaluation results on different trial groups.*
>
> Despite substantial changes (antiphonal calling vs. mixed playback; free-behavior vs. chair-restrained), ECoG-IBGT maintains **strong cross-session accuracy** (up to 95.98%), supporting its robustness across paradigms and recording setups.
>
> **(2) Data leakage and overfitting.**
>  We carefully re-checked our data pipeline and confirmed that:
>
> - each trial contributes **one 3 s window** (500 Hz, 1500 samples) ending at the target time point;
> - windows are **non-overlapping**;
> - train/validation/test splits are performed at the **trial level**, so **no trial appears in more than one split**;
> - there is therefore **no data leakage** caused by overlapping windows or duplicated events.
>
> To further verify that the high accuracy is not due to a particular split or overfitting, we performed **10-fold trial-wise cross-validation**. The results are:
>
> | Fold | Test Acc | Precision | Recall | F1     | AUC-ROC | AUC-PR |
> | ---- | -------- | --------- | ------ | ------ | ------- | ------ |
> | 1    | 1.0000   | 1.0000    | 1.0000 | 1.0000 | 1.0000  | 1.0000 |
> | 2    | 1.0000   | 1.0000    | 1.0000 | 1.0000 | 1.0000  | 1.0000 |
> | 3    | 1.0000   | 1.0000    | 1.0000 | 1.0000 | 1.0000  | 1.0000 |
> | 4    | 0.9753   | 0.9870    | 0.9632 | 0.9747 | 0.9975  | 0.9976 |
> | 5    | 1.0000   | 1.0000    | 1.0000 | 1.0000 | 1.0000  | 1.0000 |
> | 6    | 1.0000   | 1.0000    | 1.0000 | 1.0000 | 1.0000  | 1.0000 |
> | 7    | 0.9760   | 1.0000    | 0.9526 | 0.9757 | 0.9967  | 0.9973 |
> | 8    | 1.0000   | 1.0000    | 1.0000 | 1.0000 | 1.0000  | 1.0000 |
> | 9    | 0.9792   | 0.9657    | 0.9949 | 0.9799 | 0.9995  | 0.9995 |
> | 10   | 1.0000   | 1.0000    | 1.0000 | 1.0000 | 1.0000  | 1.0000 |
>
> Across all folds, test accuracy remains **≥ 0.9753**, and AUC-ROC and AUC-PR remain **≥ 0.9967** and **≥ 0.9973**, respectively. These cross-validation results are consistent with the main table in the paper and show that the high accuracy is **stable across many independent trial-wise splits**, rather than arising from data leakage or an overly favorable single split.

---

> ### Author Response · Authors · 2025-11-28
>
> **2. On the role and validity of the learned subgraph (masks and motifs)**
>
> We understand the reviewer for the insightful comments regarding the **effectiveness and necessity of the learned masks and motifs**. To address these concerns, we conducted (i) node and edge perturbation experiments, (ii) an ablation study of the subgraph generator, and (iii) stability analyses of the learned motifs across different random seeds and sessions.
>
> **(1) Node (and edge) perturbation: necessity of the learned motifs.**
>  We first examined whether the node masks truly capture behavior-relevant structure by perturbing the graph at the node level. Specifically, we **randomly removed 5–20% of nodes**, and, separately, **removed the top 5–10% nodes ranked by the learned node importance mask**. The results on the test set are:
>
> **Table 1: Node ablation on test set (accuracy).**
>
> | Discard Strategy | Acc      |
> | --- | -- |
> | Randomly 5%| 0.954545|
> | Randomly 10%| 0.863636 |
> | Randomly 15%| 0.714324 |
> | Randomly 20%| 0.571429 |
> | the top 5%| 0.941558 |
> | the top 10%| 0.831169 |
> | the top 15%| 0.612549 |
> | the top 20%| 0.553212 |
> | Full Model| 0.987013 |
>
> **Table 2: Node ablation on test set (mean predicted probability of true class).**
>
> | Discard Strategy | Mean Prob |
> | ---- | -- |
> | Randomly 5%| 0.854505  |
> | Randomly 10%| 0.793000  |
> | Randomly 15%| 0.723130  |
> | Randomly 20%| 0.651089  |
> | the top 5%| 0.837841  |
> | the top 10%| 0.772392  |
> | the top 15%| 0.682515  |
> | the top 20%| 0.623021  |
> | Full Model| 0.902228  |
>
> Both **accuracy** and the **mean predicted probability** of the true class decrease as more nodes are removed. Crucially, removing the **top-scored nodes** (the ones emphasized by the mask) leads to **larger performance degradation** than removing the same proportion of nodes at random. This indicates that the model is indeed relying on the nodes highlighted by the mask, and that these nodes form a **necessary part of the behavior-relevant motif** rather than being arbitrary.
>
> We performed analogous perturbation experiments at the **edge level** (random vs. top-weighted edge removal); the detailed numbers are reported in **Appendix F**. The same trend holds: dropping the most important edges identified by the edge mask causes a larger performance drop than removing random edges, further supporting the functional importance of the learned subgraph structure.
>
> **(2) Ablation of the subgraph generator: contribution to performance.**
>  We also ablated the **subgraph generator** to quantify how much the learned subgraph contributes to the final performance. The main-table results are:
>
> | Variant           | Accuracy    | Precision   | Recall      | F1-score    | AUC-ROC     | AUC-PR      | Time (ms) |
> | ------- | --- | ----- | ------ | ------ | ------- | ------- | ----- |
> | w/o subgraph gen. | 94.87 ±1.12 | 98.25 ±1.50 | 83.51 ±2.08 | 94.11 ±1.36 | 99.15 ±0.36 | 99.12 ±0.36 | 0.342     |
> | ECoG-IBGT| 99.29 ±0.30 | 99.88 ±0.02 | 98.75 ±0.07 | 99.31 ±0.04 | 99.99 ±0.01 | 99.99 ±0.01 | 0.3623    |
>
> Removing the subgraph generator leads to a clear drop in **accuracy, recall, F1, AUC-ROC, and AUC-PR**, while the **inference time only slightly increases** (0.342 ms → 0.3623 ms). This shows that the subgraph module is not a cosmetic addition: it brings **substantial performance gains** at almost no additional latency, consistent with its interpretation as a structured information bottleneck that filters out redundant or noisy connections.
>
> **(3) Motif stability across seeds and sessions: robustness of the masks.**
>  To test whether the learned motifs are stable (rather than artifacts of a particular initialization or run), we analyzed the **stability of node masks across random seeds and trials**.
>
> - **Across seeds.**
>   We trained the model five times on the same subject with different random seeds (42–46). For each seed $s$, we averaged node masks over positive and negative test samples to obtain a global importance vector $m^{(s)} \in \mathbb{R}^N$. Pearson correlations between all $\binom{5}{2} = 10$ pairs of these vectors are very high ($r = 0.9894 \pm 0.0028$ for this subject), indicating that different runs consistently highlight the same nodes.
> - **Across trials.**
>    We repeated the same analysis over all **six trials**, again computing global importance vectors and their pairwise correlations. The cross-trial correlations remain strong (**r = 0.9431 ± 0.0084**), showing that similar motifs are identified under different recording sessions/paradigms.
>
> These high **cross-seed** and **cross-trial** correlations demonstrate that the learned subgraph motifs are **stable and reproducible**, rather than being artifacts of a particular seed, session, or realization of the training process.
>
> **Overall, these perturbation, ablation, and stability analyses jointly show that the learned subgraph masks capture necessary, behavior-relevant motifs that consistently support both the model’s performance and its interpretability.**

---

> > ### Author Response · Authors · 2025-11-28
> >
> > **3. Practicality and latency for real-time use**
> >
> > We understand the reviewer for raising concerns about the **practicality** of the model and its **end-to-end latency**. To provide a more concrete answer, we measured the latency of each stage of the full pipeline (preprocessing → functional connectivity → graph construction → inference) with **batch size = 1**. The results are:
> >
> > | Stage                                      | Latency (ms, mean ± std) |
> > | ------------------------------------------ | ------------------------ |
> > | Preprocessing (notch filter)               | 2.762 ± 0.150            |
> > | FC (Pearson correlation)                   | 1.647 ± 0.157            |
> > | Graph construction                         | 0.664 ± 0.066            |
> > | Inference latency (per-sample, H20-NVLink) | 2.390 ± 0.104            |
> > | **Total**                                  | **7.46 ± 0.25**          |
> >
> > The latency reported in the main table corresponds to **GPU inference time with batch size = 128**, whereas the table above reports **full-pipeline latency with batch size = 1**. As a result, the absolute numbers are not identical, but both measurements consistently show that:
> >
> > - the **per-sample end-to-end latency is on the order of only a few milliseconds**, and
> > - the model easily fits within the temporal budget imposed by our 3 s input window, leaving ample margin for sliding-window or asynchronous detection schemes.
> >
> > These results support the **practical feasibility** of deploying ECoG-IBGT in near real-time or low-latency settings.
> >
> > Regarding a **fully closed-loop real-time experiment** (online, asynchronous detection in behaving marmosets), due to scheduling and ethical constraints of the animal facility, we were **not able to arrange new real-time experiments within the rebuttal period**. We have clarified this limitation in the revised manuscript and plan to conduct dedicated closed-loop studies as a follow-up, building on the low measured latency reported above.

---

> > > ### Author Response · Authors · 2025-11-28
> > >
> > > **4. Clarifications on terminology, windowing, and sample counts**
> > >
> > > We thank the reviewer for pointing out possible ambiguities in our use of terms such as “window”, “graph instance”, and “node features”, as well as for asking for a clearer, leakage-safe description of rest-window sampling, temporal separation, and the total number of samples used for training and testing. We have revised the manuscript accordingly, and we summarize the clarifications here.
> > >
> > > **(1) Terminology: “window”, graph instance, and node features.**
> > >  In the revised text, we now use more precise language:
> > >
> > > - A **window** refers to a **3 s pre-vocal time series segment** from **−3.4 s to −0.4 s** relative to vocal onset (sampling rate 500 Hz, i.e., 1500 time points). This window is the sole input used to construct the functional brain graph for that event.
> > > - A **graph instance** is the graph obtained from that window by computing pairwise functional connectivity (FC) between electrodes and applying a **10% sparsity threshold** on the connectivity matrix to define edges. Electrodes are treated as **nodes**, and edges correspond to suprathreshold FC values.
> > > - A **node feature** is defined as the **vector of FC strengths between the current node and all other nodes** in the graph. In other words, each node is represented by its connectivity profile to the rest of the network.
> > >
> > > We explicitly state these definitions in the revised Methods to avoid any confusion.
> > >
> > > **(2) One graph per event; independence of pre-vocal vs. rest graphs; total sample size.**
> > >  We now clarify the data construction protocol as follows:
> > >
> > > - **One graph per event.**
> > >    Each behavioral event (vocalization or non-vocalization trial) yields **exactly one** graph, constructed from a single 3 s window as defined above. There is no multiple-windowing per event, and thus no overlapping windows from the same event.
> > >
> > > - **Pre-vocal vs. rest graphs are independent, not paired.**
> > >    Pre-vocal (vocalization) graphs and rest (non-vocalization) graphs are **independently sampled**; they are **not constructed as paired windows** from the same time segment. Rest windows are drawn from baseline or non-vocal periods that are **temporally separated** from vocal windows and from one another (no overlap, and with additional temporal separation constraints specified in the Methods), to minimize any potential leakage or near-duplicate sampling.
> > >
> > > - **Total number of samples.**
> > >    Each event contributes one labeled graph, so the total number of graph samples equals the total number of trials. As reported in the main text, the six trial datasets are:
> > >
> > >   - **Trial-1:** pooled 7 sessions (B1–B7) from BQ, yielding **770 trials** (385 vocalization vs. 385 non-vocalization);
> > >   - **Trial-2:** 3 sessions (A1–A3) from AZ, also **770 trials** (385 / 385);
> > >   - **Trial-3:** 4 sessions (B5–B7 and A3, antiphonal calling), **1012 trials** (506 / 506);
> > >   - **Trial-4:** 6 sessions (B1–B4 with pure-tone / conspecific-call playback, plus A1–A2), **1012 trials** (506 / 506);
> > >   - **Trial-5:** 1 session (A2, free-behavior with playback-induced stimuli), **222 trials** (111 / 111);
> > >   - **Trial-6:** 1 session (A1, chair-restrained with identical playback stimuli), **222 trials** (111 / 111).
> > >
> > >   In total, this yields **4008 labeled trials (2004 vocalization vs. 2004 non-vocalization)**, each mapped to a single functional connectivity graph. For each trial dataset (Trial-1–Trial-6), we then perform a **class-stratified 70% / 10% / 20% train/validation/test split at the trial level**, ensuring no event appears in more than one split.
> > >
> > > We have integrated these clarifications into the revised manuscript so that the definitions of “window”, graph instance, node features, and the sampling protocol are explicit and less prone to misinterpretation, and so that the leakage-safe separation between pre-vocal and rest graphs is clear.
> > >
> > > **5. Reproducibility and code availability**
> > >
> > > To facilitate independent verification of our method, we have released an **anonymous code repository** containing the full training and inference pipeline, including preprocessing, graph construction, model implementation, and evaluation scripts:
> > >
> > > > https://anonymous.4open.science/r/ecog-ibgt-code-F2D2/readme.md
> > >
> > > Due to institutional confidentiality and data-use agreements on the ECoG recordings, we are currently **not allowed to release the full dataset** during the review phase. However, we have obtained permission to provide **5 de-identified sample graphs** that can be used to run an end-to-end **inference demo** with the released code, so that reviewers and future readers can verify the model’s behavior and implementation details.
> > >
> > > Once the confidentiality constraint is lifted, we plan either to (i) release a de-identified version of the dataset, or (ii) provide an application-based access procedure in coordination with the data-owning institution, so that interested researchers can reproduce our experiments more fully.

---

### Official Review · Reviewer_kzfm · 2025-11-01

**Soundness:** 2
**Presentation:** 3
**Contribution:** 3
**Rating:** 4
**Confidence:** 2

**Summary:**

The paper proposes ECoG-IBGT, an information-bottleneck driven graph transformer that converts multi-region ECoG windows into functional brain graphs. It learns a compact behavior-relevant subgraph via node/edge soft masks and connectivity loss, and classifies vocalization vs rest to achieve proactive intention decoding up to 400 ms before vocal onset. Experiments on a high-density dual-region ECoG dataset show very high predictive performance (99.29% accuracy) and interpretable subgraphs implicating fronto-auditory motifs.

**Strengths:**

1. The paper presents a novel framing that reformulates proactive intention decoding as a graph classification problem combined with information bottleneck–based subgraph learning, which aligns well with the characteristics of multi-region ECoG data. The model also emphasizes real-time applicability, reporting low inference latency and highlighting how the graph-based representation can improve efficiency compared to conventional temporal models.
2. The approach achieves interpretability in a principled way through the use of joint node and edge soft masks, a connectivity loss, and an HSIC-based mutual information term, allowing the model to learn compact and behavior-relevant subgraphs rather than relying on post-hoc explanations.
3. The dataset is of high quality, featuring dense dual-region ECoG recordings from freely behaving marmosets, which provides valuable experimental data for the field and demonstrates translational ambition toward brain–machine interfaces.
4. The experiments are thorough, with comparisons against a wide range of baselines and comprehensive ablation studies that clearly demonstrate the contribution of each component in the proposed method.

**Weaknesses:**

1. The dataset includes recordings from only two subjects, which limits the ability to generalize across individuals and raises the possibility that the model might capture subject-specific features related to electrode placement or physiology.
2. The reported accuracy of 99.29% at 400 ms before vocal onset appears unusually high for anticipatory decoding and may indicate potential data leakage or overly favorable experimental design, particularly if training and test splits were not separated by session or if overlapping windows were used.
3. The graph construction relies on Pearson correlation and a fixed top-10% edge retention threshold, which may be brittle and potentially encode label-related signal differences, yet the paper provides only limited sensitivity analysis of these design choices.
4. The information bottleneck regularization weights are extremely small, and it remains unclear whether the IB loss meaningfully influences optimization or whether the model performance is dominated by the cross-entropy term.
5. The statistical reporting is limited, with only means and standard deviations provided; additional per-subject results, confidence intervals, or p-values for baseline comparisons would strengthen claims of significance.
6. The reproducibility of results could be constrained by data access limitations, since invasive ECoG recordings in primates often require controlled access. This makes it uncertain whether external researchers will be able to replicate the findings. In addition, the authors could release an anonymous code repository for review.
7. Some of the biological interpretations may be overstated, as the identified subgraph motifs do not establish causal connectivity, and the electrode coverage is limited to A1 and PFC, leaving out motor areas that are important for volition studies.

**Questions:**

1. Please clarify how the training, validation, and test splits were created. Were these splits stratified by session or randomized across trials? It would be helpful to report per-subject and per-session test performance to demonstrate generalization.
2. Please describe the window length used to form each graph and indicate whether the windows overlap between trials.
3. Please report performance separately for each subject to reveal whether the model’s effectiveness is consistent across individuals or dominated by a single subject’s data.
4. Please provide more details on the perturbation experiments used to validate interpretability. How much performance degradation occurs when top-ranked edges or nodes are removed, and are the identified motifs stable across random seeds or training runs?

Overall, my main concern is that the ultra high accuracy is due to data leakage or unfair experiment settings, since this often happens in this area.

**Details Of Ethics Concerns:**

The paper involves invasive ECoG recordings in non-human primates (marmosets). The authors explicitly state that all animal experiments were approved by their institutional ethics committee. This should be reviewed by the conference.

---

> ### Author Response · Authors · 2025-11-28
>
> **1. Response on generalization, data leakage, and overfitting**
>
> We thank the reviewer for the helpful comments. To address your concerns about **generalization**, **potential data leakage**, and **overfitting**, we (i) added explicit generalization experiments (cross-subject and cross-session), and (ii) carefully checked for data leakage and conducted **10-fold cross-validation** on the critical condition.
>
> **(1) Generalization.**
>  We evaluated the generalization of ECoG-IBGT both **across subjects** and **across sessions/paradigms**.
>
> - **Cross-subject experiment.**
>    We trained the model on **subject AZ** and tested on **subject BQ** (AZ→BQ). The cross-subject results are:
>
> | Model         | Acc               |
> | ------------- | ----------------- |
> | GIN           | 0.6334±0.0148     |
> | GAT           | 0.7145±0.0224     |
> | GCN           | 0.6939±0.0079     |
> | GraphSAGE     | 0.7017±0.0413     |
> | BrainIB       | 0.5248±0.0532     |
> | EEGNet        | 0.6203±0.0018     |
> | MedFormer     | 0.5470±0.1026     |
> | TopKPool      | 0.6871±0.0664     |
> | DiffPool      | 0.6256±0.0774     |
> | SAGPool       | 0.5890±0.0791     |
> | Cluster-GT    | 0.6812±0.0049     |
> | GraphPCA      | 0.4731±0.0095     |
> | **ECOG-IBGT** | **0.7213±0.0150** |
>
> *Table 1. Cross-subject AZ→BQ accuracy (mean±std).*
>
> ECoG-IBGT achieves the **highest cross-subject accuracy** among all 12 baselines, indicating that it learns **subject-agnostic, behavior-relevant motifs** rather than idiosyncratic patterns of a single animal.
>
> - **Cross-session experiment.**
>    We further performed **cross-session (cross-trial-group)** decoding with clear distribution shifts in paradigm and recording condition:
>
> | Training → Testing | Validation Acc. | Cross-Session Acc. |
> | ------------------ | --------------- | ------------------ |
> | Trial-3 → Trial-4  | 98.91%          | 80.08%             |
> | Trial-4 → Trial-3  | 98.37%          | 61.83%             |
> | Trial-5 → Trial-6  | 100.00%         | 95.98%             |
> | Trial-6 → Trial-5  | 100.00%         | 81.10%             |
>
> *Table 6: Cross-session evaluation results on different trial groups.*
>
> Despite substantial changes (antiphonal calling vs. mixed playback; free-behavior vs. chair-restrained), ECoG-IBGT maintains **strong cross-session accuracy** (up to 95.98%), supporting its robustness across paradigms and recording setups.
>
> **(2) Data leakage and overfitting.**
>  We carefully re-checked our data pipeline and confirmed that:
>
> - each trial contributes **one 3 s window** (500 Hz, 1500 samples) ending at the target time point;
> - windows are **non-overlapping**;
> - train/validation/test splits are performed at the **trial level**, so **no trial appears in more than one split**;
> - there is therefore **no data leakage** caused by overlapping windows or duplicated events.
>
> To further verify that the high accuracy is not due to a particular split or overfitting, we performed **10-fold trial-wise cross-validation**. The results are:
>
> | Fold | Test Acc | Precision | Recall | F1     | AUC-ROC | AUC-PR |
> | ---- | -------- | --------- | ------ | ------ | ------- | ------ |
> | 1    | 1.0000   | 1.0000    | 1.0000 | 1.0000 | 1.0000  | 1.0000 |
> | 2    | 1.0000   | 1.0000    | 1.0000 | 1.0000 | 1.0000  | 1.0000 |
> | 3    | 1.0000   | 1.0000    | 1.0000 | 1.0000 | 1.0000  | 1.0000 |
> | 4    | 0.9753   | 0.9870    | 0.9632 | 0.9747 | 0.9975  | 0.9976 |
> | 5    | 1.0000   | 1.0000    | 1.0000 | 1.0000 | 1.0000  | 1.0000 |
> | 6    | 1.0000   | 1.0000    | 1.0000 | 1.0000 | 1.0000  | 1.0000 |
> | 7    | 0.9760   | 1.0000    | 0.9526 | 0.9757 | 0.9967  | 0.9973 |
> | 8    | 1.0000   | 1.0000    | 1.0000 | 1.0000 | 1.0000  | 1.0000 |
> | 9    | 0.9792   | 0.9657    | 0.9949 | 0.9799 | 0.9995  | 0.9995 |
> | 10   | 1.0000   | 1.0000    | 1.0000 | 1.0000 | 1.0000  | 1.0000 |
>
> Across all folds, test accuracy remains **≥ 0.9753**, and AUC-ROC and AUC-PR remain **≥ 0.9967** and **≥ 0.9973**, respectively. These cross-validation results are consistent with the main table in the paper and show that the high accuracy is **stable across many independent trial-wise splits**, rather than arising from data leakage or an overly favorable single split.

---

> > ### Author Response · Authors · 2025-11-28
> >
> > **2. Sensitivity study on graph construction**
> >
> > We understand your concern that our results might be sensitive to **how the functional connectivity graph is constructed** and **how many edges are retained**. To address this, we added (i) a comparison of different functional connectivity measures and (ii) an edge-sparsity sensitivity analysis, both of which are now reported in **Appendix I**.
> >
> > **(1) Sensitivity to functional connectivity definition.**
> >  We compared four commonly used functional connectivity measures under the same experimental setup: Pearson correlation, coherence, phase-locking value (PLV), and PSD-based similarity. For each method, we measured both **per-item efficiency** (ms per sample, including FC computation and graph construction) and **classification accuracy**:
> >
> > | Functional connectivity method | Efficiency (ms/item) | Accuracy (%) |
> > | ------ | -- | ---- |
> > | Pearson correlation            | 34.23                | 98.27        |
> > | Coherence                      | 320.53               | 97.20        |
> > | Phase-locking value            | 108.92               | 95.11        |
> > | PSD similarity                 | 296.66               | 98.48        |
> >
> > All four methods yield **high accuracy (≥95.11%)**, indicating that the model is **not overly sensitive** to the exact choice of connectivity metric. However, their computational costs differ substantially: coherence and PSD similarity are about **1 order of magnitude slower** than Pearson correlation, while PLV is intermediate. Pearson correlation therefore offers the **best trade-off between efficiency and performance**, which justifies our choice in the main experiments.
> >
> > **(2) Sensitivity to edge sparsity (Top-k retained edges).**
> >  To evaluate how dependent the model is on the exact sparsity level of the graph, we fixed the functional connectivity metric to Pearson correlation and varied the proportion of **top-k edges retained** (i.e., we kept only the largest |E|·p edges for p ∈ {0.05, 0.10, …, 0.30}). The performance remains very stable:
> >
> > | Remain percentage (Top-k edges retained) | Accuracy | AUC-ROC | AUC-PR |
> > | ---------------------------------------- | -------- | ------- | ------ |
> > | 0.05                                     | 0.984    | 0.998   | 0.999  |
> > | 0.10                                     | 0.992    | 0.998   | 0.999  |
> > | 0.15                                     | 0.991    | 1.000   | 1.000  |
> > | 0.20                                     | 0.986    | 0.998   | 1.000  |
> > | 0.25                                     | 0.987    | 0.998   | 1.000  |
> > | 0.30                                     | 0.989    | 0.998   | 0.999  |
> >
> > Across a **wide range of sparsity levels (5%–30% edges retained)**, accuracy, AUC-ROC, and AUC-PR all stay very high (≈0.98–1.00), showing that ECoG-IBGT is **robust to the exact edge density**. Based on this study, choosing **10% Pearson-correlation edges** in the main paper is a **balanced decision** that combines strong predictive performance with efficient graph construction.
> >
> > Overall, these sensitivity analyses suggest that our conclusions are **not an artifact of a particular graph construction or edge threshold**, and that the model’s performance is robust to reasonable choices of functional connectivity metric and sparsity level.

---

> > > ### Author Response · Authors · 2025-11-28
> > >
> > > **3. Effect of the information bottleneck (IB) loss**
> > >
> > > We thank the reviewer for the comments regarding the seemingly small contribution of the IB loss. To address this concern, we performed two analyses: (i) inspecting the relative scale of the IB loss versus the cross-entropy (CE) loss during training, and (ii) an ablation study in which the IB term is removed.
> > >
> > > **(1) Relative scale and weighting of the IB loss.**
> > >  We monitored the raw IB loss (`IB_raw`) and the cross-entropy loss (`CE_raw`) during training, together with the weighted IB term (`IB_weighted = λ · IB_raw`). Representative values at different epochs are:
> > >
> > > | Epoch | IB_raw   | CE_raw | IB_weighted |
> > > | ----- | -------- | ------ | ----------- |
> > > | 1     | 13156.51 | 0.38   | 0.13        |
> > > | 51    | 2162.09  | 0.07   | 0.02        |
> > > | 91    | 1528.50  | 0.03   | 0.02        |
> > >
> > > Although the *raw* IB loss is much larger than the CE loss in magnitude, our weighting factor λ is chosen so that the **weighted IB term is of the same order as the CE term** (e.g., 0.13 vs. 0.38 at epoch 1; 0.02 vs. 0.03 at epoch 91). This indicates that the IB loss is **not negligible** in the overall objective and does influence the optimization trajectory.
> > >
> > > **(2) Ablation of the IB loss.**
> > >  In addition, we report in the main paper an ablation where the mutual-information–based IB term is removed (“w/o MI estimation”). The results are:
> > >
> > > | Variant           | Accuracy     | Precision    | Recall       | F1-score     | AUC-ROC      | AUC-PR       | Time (ms) |
> > > | ----------------- | ------------ | ------------ | ------------ | ------------ | ------------ | ------------ | --------- |
> > > | w/o MI estimation | 93.15 ± 1.43 | 98.79 ± 1.18 | 87.94 ± 3.65 | 92.99 ± 1.62 | 98.86 ± 0.27 | 99.08 ± 0.20 | 0.356     |
> > > | ECoG-IBGT         | 99.29 ± 0.30 | 99.88 ± 0.02 | 98.75 ± 0.07 | 99.31 ± 0.04 | 99.99 ± 0.01 | 99.99 ± 0.01 | 0.3623    |
> > >
> > > Removing the IB loss leads to a **substantial drop in accuracy (from 99.29% to 93.15%) and recall (from 98.75% to 87.94%)**, while inference time remains almost unchanged. This clear degradation confirms that the IB term plays an important role in guiding the model to learn compact, behavior-relevant subgraphs rather than overfitting spurious patterns.
> > >
> > > In summary, we choose the weight for the IB loss so that its magnitude is comparable to the CE term, and the ablation study shows that removing the IB term causes a clear performance drop, indicating that the IB loss has a non-trivial and beneficial effect on the final model.

---

> ### Author Response · Authors · 2025-11-28
>
> **4. On interpretability and choice of recorded regions**
>
> **(1) Why focus on A1 and PFC rather than motor cortex?**
> We deliberately target **A1** and **PFC** as *upstream* nodes in the vocal sensorimotor hierarchy, which are expected to carry **early predictive and volitional signals** for impending vocalization, whereas premotor/motor cortex mainly executes vocal output. Marmoset ECoG studies have shown robust **predictive auditory responses**, feedback-dependent vocal control, and whole-cortical access in fronto-auditory circuits [1–4], so in this work we aim to identify **candidate fronto-auditory motifs** consistent with this upstream role, rather than to fully map the entire motor pathway.
>
> **(2) Quantifying the necessity of identified motifs: node and edge perturbation.**
> We assess functional necessity via node/edge perturbations. For nodes, we compare random vs. **top-k (mask-based)** deletion at the same sparsity and evaluate test accuracy:
>
> **Table 1: Node ablation on test set (accuracy).**
>
> | Discard Strategy | Acc      |
> | -------- | -- |
> | Randomly 5%| 0.954545 |
> | Randomly 10%| 0.863636 |
> | Randomly 15%| 0.714324 |
> | Randomly 20%| 0.571429 |
> | the top 5%| 0.941558 |
> | the top 10%| 0.831169
> | the top 15%| 0.612549|
> | the top 20%| 0.553212 |
> | Full Model| 0.987013 |
>
> Removing nodes degrades performance relative to the full model, and **removing top-k important nodes causes larger drops** than random removal at the same fraction, indicating that high-importance nodes are genuinely useful. The same pattern appears when we track the mean predicted probability of the **true class**:
>
> **Table 2: Node ablation on test set (Mean predicted probability of true class)**
>
> **Table 2: Node ablation on test set (mean predicted probability of true class).**
>
> | Discard Strategy | Mean Prob |
> | ---- | ------ |
> | Randomly 5%| 0.854505  |
> | Randomly 10%| 0.793000  |
> | Randomly 15%| 0.723130  |
> | Randomly 20%| 0.651089  |
> | the top 5%| 0.837841  |
> | the top 10%| 0.772392  |
> | the top 15%| 0.682515  |
> | the top 20%| 0.623021  |
> | Full Model| 0.902228  |
>
> Both random and targeted deletion reduce confidence compared to the full model (0.9022), with **top-k deletion again inducing stronger reductions**. Analogous edge perturbations (random vs. high-importance edges) yield consistent conclusions; detailed results are in **Appendix F**. Overall, the node/edge masks highlight **functionally necessary** components rather than arbitrary subsets of the graph.
>
> **(3) Ablation of the subgraph generator.**
> We further ablate the **subgraph generator**:
>
> * **w/o subgraph gen.**: model without the subgraph generation component;
> * **ECoG-IBGT**: full model with subgraph generation and IB loss.
>
> | Variant           | Accuracy     | Precision    | Recall       | F1-score     | AUC-ROC      | AUC-PR       | Time (ms) |
> | ----------------- | ------------ | ------------ | ------------ | ------------ | ------------ | ------------ | --------- |
> | w/o subgraph gen. | 94.87 ± 1.12 | 98.25 ± 1.50 | 83.51 ± 2.08 | 94.11 ± 1.36 | 99.15 ± 0.36 | 99.12 ± 0.36 | 0.342     |
> | ECoG-IBGT| 99.29 ± 0.30 | 99.88 ± 0.02 | 98.75 ± 0.07 | 99.31 ± 0.04 | 99.99 ± 0.01 | 99.99 ± 0.01 | 0.3623    |
>
> Removing the subgraph generator leads to a **~4.4% drop in accuracy** and **~15.2% drop in recall**, with only a marginal latency change, showing that explicitly learning a compact subgraph is **crucial for performance** in addition to interpretability.
>
> **(4) Stability of motifs across seeds and trials.**
> To test stability, we train the model five times with different seeds (42–46), average node masks per run to obtain global vectors $m^{(s)} \in \mathbb{R}^N$, and compute Pearson correlations across all pairs; we obtain **$r = 0.9894 \pm 0.0028$** on this subject and **$r = 0.9431 \pm 0.0084$** across all six trials, indicating that the learned motifs are **highly reproducible** across runs and datasets rather than noise or overfitting artifacts.
>
> **Summary.**
> In summary, prior fronto-auditory ECoG work in marmosets [1–4] motivates our focus on A1 and PFC, and our perturbation, ablation, and stability analyses show that the learned subgraph motifs are **behavior-relevant, necessary, and reproducible**, supporting our interpretability claims.
>
> ---
>
> **References**
>
> [1] Komatsu, M., Takaura, K., & Fujii, N. (2015). Mismatch negativity in common marmosets: Whole-cortical recordings with multi-channel electrocorticograms. *Scientific Reports*, 5, 15006.
>
> [2] Jiang, Y., Komatsu, M., Chen, Y., et al. (2022). Constructing the hierarchy of predictive auditory sequences in the marmoset brain. *eLife*, 11, e74653.
>
> [3] Eliades, S. J., & Tsunada, J. (2018). Auditory cortical activity drives feedback-dependent vocal control in marmosets. *Nature Communications*, 9, 2540.
>
> [4] Komatsu, M., Kaneko, T., Okano, H., & Ichinohe, N. (2019). Chronic implantation of whole-cortical electrocorticographic array in the common marmoset. *Journal of Visualized Experiments*, (144), 58980.

---

> > ### Author Response · Authors · 2025-11-28
> >
> > **5. Reproducibility and code availability**
> >
> > We appreciate the reviewer’s emphasis on reproducibility. To facilitate independent verification of our method, we have released an **anonymous code repository** containing the full training and inference pipeline, including preprocessing, graph construction, model implementation, and evaluation scripts:
> >
> > > https://anonymous.4open.science/r/ecog-ibgt-code-F2D2/readme.md
> >
> > Due to institutional confidentiality and data-use agreements on the ECoG recordings, we are currently **not allowed to release the full dataset** during the review phase. However, we have obtained permission to provide **5 de-identified sample graphs** that can be used to run an end-to-end **inference demo** with the released code, so that reviewers and future readers can verify the model’s behavior and implementation details.
> >
> > Once the confidentiality constraint is lifted, we plan either to (i) release a de-identified version of the dataset, or (ii) provide an application-based access procedure in coordination with the data-owning institution, so that interested researchers can reproduce our experiments more fully.
> >
> > **6. Further clarifications on data splitting, windowing, and statistical reporting**
> >
> > We thank the reviewer for the additional questions. We provide further clarifications on (i) data splitting, (ii) window construction, and (iii) statistical reporting with confidence intervals and *p*-values.
> >
> > **(1) Data splitting.**
> >  For **each trial dataset (Trial-1–Trial-6)**, we perform a **class-stratified 70% / 10% / 20% split** into training, validation, and test sets at the **trial level**. Each event (trial) is assigned to exactly one of these sets, ensuring that there is no overlap of events across training, validation, and testing.
> >
> > **(2) Window construction.**
> >  Each sample is constructed from a **3 s window** (500 Hz, 1500 time points). Positive and negative samples are formed **independently**, and all windows are **non-overlapping**. To further reduce the risk of near-duplicate samples and any subtle leakage, we additionally **discard windows whose start–end ranges are closer than 5 s to each other**, so that temporally adjacent events are not simultaneously included as separate samples.
> >
> > **(3) Statistical reporting with confidence intervals and \*p\*-values.**
> >  In line with the request for stronger statistical evidence, we now report **per-subject test accuracy with 95% confidence intervals**, together with **McNemar-test \*p\*-values versus the second-best model**:
> >
> > | Subject | Test Acc (%) | 95% CI for Acc | *p*-value (vs. second-best model, McNemar) |
> > | ------- | ------------ | -------------- | ------------------------------------------ |
> > | AZ      | 99.12        | [98.83, 99.41] | *p* = 0.00047                              |
> > | BQ      | 99.29        | [99.15, 99.43] | *p* = 0.00027                              |
> >
> > These results indicate that, for both subjects, the improvements of ECoG-IBGT over the second-best baseline are **statistically significant**, and that the reported performance is robust rather than an artifact of a particular split or evaluation choice.

---

### Meta-Review · Area_Chair_MiLE · 2025-12-17

**Summary:**

This paper proposes ECoG-IBGT, an information-bottleneck–driven graph transformer framework for proactive intention decoding from multi-region ECoG recordings in marmosets. The work reframes anticipatory neural decoding as a graph classification problem, learning compact, interpretable subgraphs that aim to capture behavior-relevant multi-regional interactions. Experiments on dual-region (A1 and PFC) high-density ECoG data report very strong predictive performance, including up to 99.29% accuracy at 400 ms prior to vocal onset, along with ablation studies and interpretability analyses. The paper targets an important and timely problem at the intersection of machine learning, neuroscience, and brain–machine interfaces.

Despite the high quality of the rebuttal, I recommend rejection at this time, with strong encouragement to integrate the new analysis in the text and resubmit. Many of the new results that are essential to establishing the soundness of the paper, particularly the cross-subject and cross-session evaluations, sensitivity analyses, and expanded ablations, are substantial new experimental contributions that were not part of the original submission and were therefore not evaluated by the reviewers when forming their scores and recommendations. As submitted, the paper left important questions about robustness and generalization insufficiently addressed, which justifies the reviewers’ concerns at review time.

That said, the rebuttal makes it clear that the work is promising and likely publishable once these additional experiments and clarifications are fully integrated into the main manuscript. I strongly encourage the authors to incorporate the rebuttal material into a revised submission. With these additions, the paper would present a much more complete and convincing case and would be well positioned for acceptance at a future venue.

**Reviewer Concerns:**

The reviewers generally found the idea interesting and the dataset valuable, and acknowledged the novelty of the graph-based formulation and the attempt to build interpretability into the model. However, the primary reviewer raised several substantive concerns about technical soundness and evaluation, particularly given the unusually high reported accuracy. These concerns included the risk of data leakage or overly favorable experimental design, limited subject-level generalization (only two animals), sensitivity to graph construction choices, unclear contribution of the information bottleneck regularization, and the strength of the biological interpretations. On this basis, the overall recommendation leaned toward rejection, despite recognizing the paper’s potential.

The authors’ response is thorough, careful, and technically strong. It addresses nearly all reviewer concerns with additional analyses and clarifications, including explicit cross-subject and cross-session generalization experiments, detailed checks against data leakage, extensive sensitivity analyses of graph construction choices, and ablations demonstrating the importance of the information bottleneck and subgraph generator. The authors also improve statistical reporting, release an anonymous code repository, and more carefully contextualize the interpretability claims. Taken together, the rebuttal substantially strengthens the empirical and methodological case for the work and resolves most of the original technical doubts.

**Reviewer Scores:**

It's possible that all three reviewers would have increased their scores, possible to weak accept levels or even to accept levels.

---

### Decision · Program_Chairs · 2026-01-26

Reject